# Structural basis for botulinum neurotoxin E recognition of synaptic vesicle protein 2

Zheng Liu [1], Pyung-Gang Lee[2,3], Nadja Krez [4], Kwok-ho Lam[1], Hao Liu[2,3], Adina Przykopanski[4], Peng Chen [1], Guorui Yao[1], Sicai Zhang[2,3], Jacqueline M. Tremblay[5], Kay Perry [6], Charles B. Shoemaker [5], Andreas Rummel [4], Min Dong [2,3] ✉ & Rongsheng Jin [1] ✉

Botulinum neurotoxin E (BoNT/E) is one of the major causes of human botulism and paradoxically also a promising therapeutic agent. Here we determined the co-crystal structures of the receptor-binding domain of BoNT/E ($H_C$E) in complex with its neuronal receptor synaptic vesicle glycoprotein 2A (SV2A) and a nanobody that serves as a ganglioside surrogate. These structures reveal that the protein-protein interactions between $H_C$E and SV2 provide the crucial location and specificity information for $H_C$E to recognize SV2A and SV2B, but not the closely related SV2C. At the same time, $H_C$E exploits a separated sialic acid-binding pocket to mediate recognition of an N-glycan of SV2. Structure-based mutagenesis and functional studies demonstrate that both the protein-protein and protein-glycan associations are essential for SV2A-mediated cell entry of BoNT/E and for its potent neurotoxicity. Our studies establish the structural basis to understand the receptor-specificity of BoNT/E and to engineer BoNT/E variants for new clinical applications.

Botulinum neurotoxins (BoNTs) are the causative agents of the neuroparalytic disease botulism[1,2]. There are seven major BoNT serotypes (termed BoNT/A–G), among which BoNT/A and BoNT/B are approved by the FDA for clinical and esthetic indications[3,4]. The extraordinary potency of BoNTs relies on highly specific recognition and uptake by motor neurons[5,6]. A widely accepted dual-receptor model suggests that the receptor-binding domain ($H_C$) of BoNTs synergistically binds complex gangliosides and specific protein receptors on the neuron surface[7–10]. Complex gangliosides are present abundantly on nerve cells and serve to enrich toxins, and most BoNTs possess a highly conserved ganglioside-binding "SxWY" motif[11]. However, BoNTs have developed diverse binding strategies for their corresponding protein receptors. For example, BoNT/A and BoNT/B exploit distinct protein receptors[12–17], which may contribute to their differences in pharmacological and clinical profiles. BoNT/E

has recently emerged as a promising new drug candidate due to its faster onset of action and shorter duration of effect when compared to BoNT/A and BoNT/B[18–24].

At the molecular level, BoNT/E recognizes synaptic vesicle glycoprotein 2 (SV2)[25,26], a family of 12-transmembrane domain proteins that also serve as receptors for BoNT/A[12,17], BoNT/D[27], BoNT/F[28,29], and the related tetanus neurotoxin[30]. SV2 comprises three homologous isoforms, SV2A, 2B, and 2C, in mammals. Despite their similar primary sequences (~60% identity), only SV2A and SV2B, but not SV2C, are able to mediate the cell entry of BoNT/E into cultured hippocampal and cortical neurons[25], although it remains to be validated whether SV2C in motor neurons may still function as a receptor for BoNT/E[31–33]. This is in sharp contrast to BoNT/A, which is able to use all three SV2 isoforms for cell entry[12]. As the three SV2 isoforms have different tissue distributions in human[33–35], the different specificities toward SV2 isoforms

[1]Department of Physiology and Biophysics, University of California, Irvine, Irvine, CA 92697, USA. [2]Department of Urology, Boston Children's Hospital, Boston, MA 02115, USA. [3]Department of Microbiology and Department of Surgery, Harvard Medical School, Boston, MA 02115, USA. [4]Institute of Toxicology, Hannover Medical School, Hannover 30623, Germany. [5]Tufts Cummings School of Veterinary Medicine, North Grafton, MA 01536, USA. [6]NE-CAT and Department of Chemistry and Chemical Biology, Cornell University, Argonne National Laboratory, Argonne, IL 60439, USA. ✉e-mail: Min.Dong@childrens.harvard.edu; r.jin@uci.edu

between BoNT/E and BoNT/A can contribute to potential differences in pharmacological and therapeutic features.

How BoNT/E manages to distinguish SV2C from SV2A and SV2B remains a mystery. Prior studies revealed that the receptor-binding domain of BoNT/A ($H_C$A) recognizes the open edge of the most C-terminal β-strand of the quadrilateral β-helix fold of SV2C luminal domain as well as the core saccharides of a neighboring N-glycan of SV2C, which together form a composite binding site for $H_C$A[13,14,36–38]. However, the receptor-binding domain of BoNT/E ($H_C$E) has an 8-amino acid deletion and many substitutions at the homologous $H_C$A-like SV2-binding site (Supplementary Fig. 1a), suggesting that BoNT/E exploits a distinct yet unknown mechanism to recognize SV2A and SV2B.

In this study, we designed and characterized a fusion protein composed of the luminal domain of human SV2A and SV2C that maintains an SV2A-like binding capacity to BoNT/E (named SV2Ac). We also designed a fusion protein consisting of SV2Ac and a single-domain camelid antibody (a.k.a. VHH or nanobody, named G6) that binds to the ganglioside-binding site on $H_C$E and acts as a ganglioside surrogate to enhance $H_C$E–SV2A association. We then determined two crystal structures of $H_C$E in complex with SV2Ac–VHH and SV2Ac–VHH plus sialic acid. These structures reveal that BoNT/E simultaneously recognizes both specific protein segments and an N-glycan of SV2A at two separated $H_C$E sites. Complementary biophysical, cellular, and functional studies demonstrated that BoNT/E specifically recognizes SV2A and SV2B, but not SV2C, via the protein–protein interface, while it also grips the tip of the SV2 glycan at a distant site that strengthens the association. Both of the protein–protein and protein–glycan-binding modes between BoNT/E and SV2A are distinct from that between BoNT/A and SV2C[13,14]. These findings provide the structural basis to facilitate the therapeutic development and engineering of BoNT/E for novel neurotoxin products, as well as to inform new strategies for developing BoNT inhibitors.

## Results

### VHH-G6 blocks ganglioside binding of BoNT/E

We expressed and purified the fourth luminal domain of human SV2A (residues F487–E581, referred to as SV2A-L4), which we previously identified as the BoNT/E-binding fragment[25,26], as a secreted and glycosylated protein from human embryonic kidney 293 cells (HEK293)[13,39]. However, $H_C$E was found to poorly bind SV2A-L4 and their interaction was barely detectable using a pull-down assay, making it unsuitable for structural studies. We hypothesized that the low affinity between $H_C$E and SV2A-L4 was due to the lack of co-receptor gangliosides that are known to be essential for the cell entry of BoNT/E[25,26,28]. Complex gangliosides are present abundantly on nerve cell surfaces and serve to enrich toxins during the crucial early stage of cell binding. As BoNT/E-receptor recognition on cell surfaces relies on two receptors, we set out to explore a strategy to enhance SV2A binding to $H_C$E by conjugating SV2A-L4 with a ganglioside-mimicking component. Such an engineered protein should then bind $H_C$E in a manner resembling the dual receptor recognition of BoNT/E. To this end, we sought to identify a VHH that recognizes the ganglioside-binding site on BoNT/E as a surrogate for gangliosides.

We have developed numerous BoNT-binding VHHs as reagents and countermeasures, especially for BoNT/A, B, and E which cause the majority of human intoxications[40–45]. Among many BoNT/E-targeting VHHs, we focused on VHH-JLE-G6 (referred to as G6), which neutralizes BoNT/E toxicity and displays high affinity binding to $H_C$E[42]. We found that G6 markedly reduced the binding of $H_C$E to liposomes containing complex ganglioside GT1b in a co-sedimentation assay, suggesting that G6 likely competes with GT1b for $H_C$E binding (Fig. 1a). To better

understand the neutralizing mechanism of G6, we determined the co-crystal structure of an $H_C$E–G6 complex at 3.23 Å resolution (Fig. 1b and Supplementary Table 1). This structure reveals that the complementarity-determining region 3 (CDR3) of G6 forms extensive interactions with the C-terminal subdomain of $H_C$E ($H_{CC}$E) via a network of hydrogen bonds complemented with salt bridges and hydrophobic interactions, while the CDR1 and CDR2 do not directly bind $H_C$E (Fig. 1c). Since the ganglioside-binding modes are highly conserved among different BoNT serotypes, we did structural modeling based on the published structures of $H_C$A and $H_C$B in complex with gangliosides and found that G6 residues V104 and L102 bind to $H_C$E at sites that should otherwise accommodate Gal4 and Sia5 of GT1b[9,11,46,47]. As a result, $H_C$E residues W1224 and Y1225, which are part of the highly conserved ganglioside-binding "SxWY" motif, are blocked from binding gangliosides (Supplementary Fig. 1b). This finding is also consistent with the structure of $H_C$E–ganglioside complex that was published during the preparation of this manuscript (Fig. 1d)[48]. Furthermore, we previously reported that mutating $H_C$E-W1224 was sufficient to abolish its ganglioside binding[26,28]. These results demonstrate that G6 occupies the ganglioside-binding site on $H_C$E and blocks ganglioside binding, leading to BoNT/E neutralization.

### SV2A–G6 fusion protein mimics the dual receptors of BoNT/E

Earlier studies focusing on $H_C$A and $H_C$B demonstrate that the binding sites for their protein receptors and gangliosides are located in two separated but neighboring areas on $H_C$ (Supplementary Fig. 1c, d)[13–16,46,47]. We found that the structure of the G6-bound $H_C$E is virtually identical to that in complex with GD1a (root mean square deviation, r.m.s.d. ~0.38 Å over 359 aligned Cα pairs)[48]. Therefore, we hypothesized that G6 could be used as a ganglioside surrogate to facilitate SV2A binding to $H_C$E when G6 and SV2A are properly connected with a flexible peptide linker because such a fusion protein would allow synergistic binding of G6 and SV2A to $H_C$E in a way resembling the dual receptor binding. Guided by the structure of the $H_C$E–G6 complex, we designed a fusion protein in which G6 (residues Q1–S129) was linked to the C-terminus of SV2A-L4 because the N-terminus of G6 is closer to $H_C$E than its C-terminus (Fig. 1e). We employed a 10-amino acid flexible linker that should have sufficient length for SV2A-L4 to sample a large area on $H_C$E surface for binding. To validate this design, we first designed a mutated G6 that carries double mutations D100A/D115A on its CDR3 (termed G6$^{AA}$) to drastically weaken its binding to $H_C$E (Fig. 1f). We rationalized that a properly designed SV2A–G6$^{AA}$ fusion protein that structurally allows the simultaneous binding of both weak binders would display substantially enhanced avidity due to the bivalent binding, while an improperly designed fusion protein in which only one component could bind would display poor affinity. Using a pull-down assay, we found that the glycosylated SV2A–G6$^{AA}$ expressed in HEK293 cells strongly interacted with $H_C$E (Fig. 1g), suggesting that SV2A–G6$^{AA}$ successfully mimics the dual-receptor binding to $H_C$E in vitro. Interestingly, we observed that SV2A–G6$^{AA}$ robustly bound $H_C$E at neutral pH (e.g. 7.5), but not at acidic pH (e.g. 4.6 and 5.0) (Fig. 1g), which is similar to BoNT/A binding of SV2C[14,49] but different from the pH-independent binding between BoNT/B and its receptor synaptotagmin[15].

### Engineering a SV2A-SV2C chimera capable of binding to $H_C$E

We carried out systematic screenings of co-crystallization of $H_C$E in complex with SV2A–G6 in which the wild-type (WT) G6 was used to further enhance complex stability. However, despite extensive efforts, we were unable to obtain high-quality crystals for diffraction studies, which we identified was largely due to the tendency of SV2A-L4 to aggregate in solution. Interestingly, the recombinant SV2C-L4 is

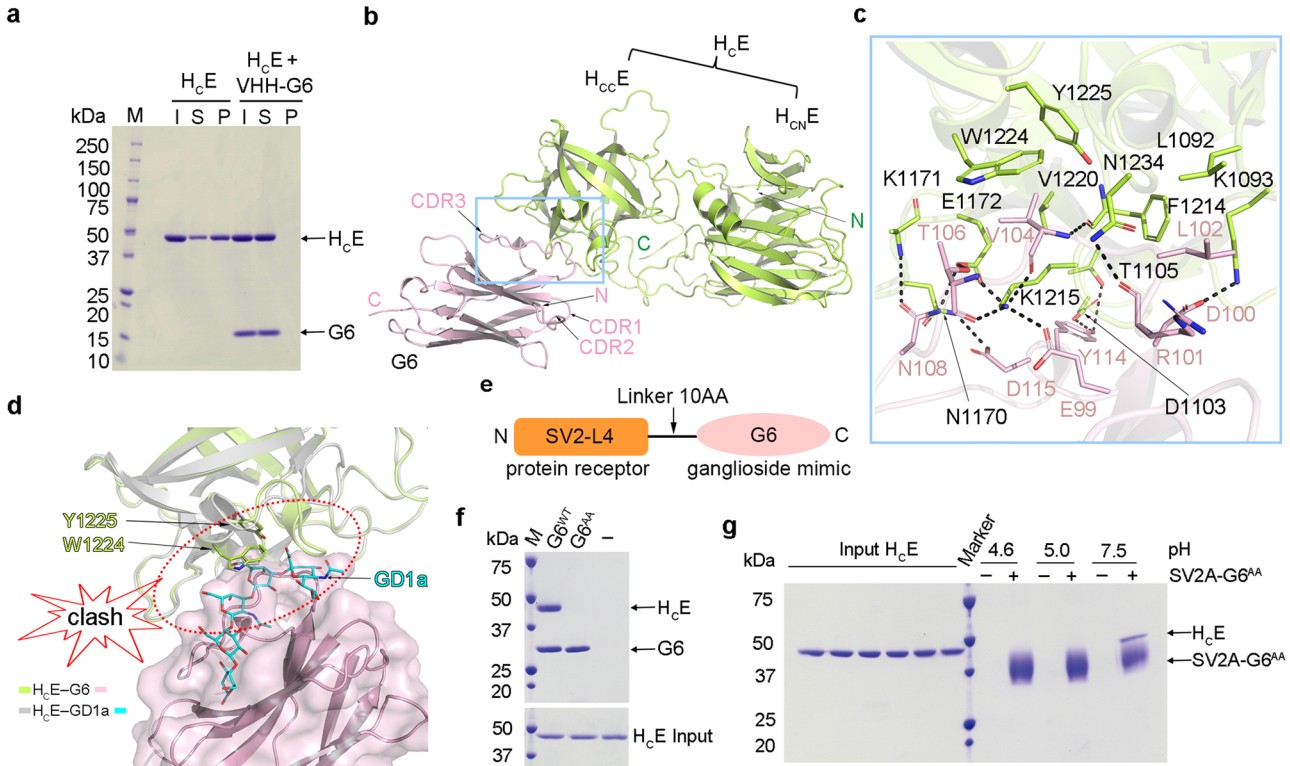

**Fig. 1 | Structure of the $H_CE$–G6 complex and the designs of SV2A–G6 fusion protein. a** $H_CE$ pre-incubated with or without G6 was incubated with liposomes containing 70/20/10 mol% BrPC/DOPS/GT1b. After liposomes were pelleted, $H_CE$ and G6 in the input (I), supernatant (S), and pellet (P) fractions were analyzed by SDS–PAGE and Coomassie blue staining (a representative result is shown, $n=3$). **b** Cartoon representation of the $H_CE$–G6 complex with $H_CE$ colored in lemon and G6 in pink. The N- and C-terminal subdomains of $H_CE$ are referred to as $H_{CN}E$ and $H_{CC}E$, respectively. **c** A close-up view of the interface between $H_CE$ and G6 is highlighted in the blue box in (**b**). Key interacting residues are shown as sticks. **d** G6 occupies the ganglioside-binding pocket on $H_CE$. The G6-bound $H_CE$ (lemon

cartoon) is superimposed with the GD1a-bound $H_CE$ (gray cartoon) (PDB: 7OVW). G6 and GD1a are shown as a pink surface model and a cyan stick model, respectively, and two crucial ganglioside-binding residues W1224 and Y1225 are shown as sticks. **e** A schematic diagram showing the design of an SV2A–G6 fusion protein. **f** The $G6^{AA}$ mutant (D100A/D115A) showed no detectable binding to $H_CE$ in a pull-down assay with $H_CE$ as prey and the His/SUMO-tagged $G6^{WT}$ or $G6^{AA}$ as bait. **g** $H_CE$ recognizes SV2A in a pH-dependent manner. Biotin-labeled $SV2A-G6^{AA}$ as bait could pull down $H_CE$ at pH 7.5, but not pH 4.6 or 5.0. Representative results are shown in panels (**a**), (**f**), and (**g**) ($n=3$).

mono-dispersed and has excellent biochemical behavior[13]. As SV2A-L4 and SV2C-L4 are homologous to each other, we sought to develop an SV2A–SV2C chimera that has improved biochemical behavior over SV2A-L4 while maintaining the SV2A-like binding with $H_CE$. To this end, we designed a series of SV2A–SV2C chimeras in the context of the SV2A–G6$^{AA}$ fusion protein by swapping fragments of SV2A and SV2C, expressed them in HEK293 cells, and then examined their biochemical features and interaction with $H_CE$ (Fig. 2a). We found that a chimera composed of the N-terminal segment of SV2C-L4 (V473–K518) and the C-terminal segment of SV2A-L4 (E533–E581) maintained SV2A-like binding to $H_CE$ based on the pull-down assay (termed SV2Ac–G6$^{AA}$), and it was mono-dispersed in solution (Fig. 2b, c). In comparison, another chimera composed of residues V473–I538 of SV2C and N553–E581 of SV2A (SV2Ac$^1$–G6$^{AA}$), as well as the stand-alone SV2C, SV2Ac, SV2Ac$^1$, SV2A, or SV2C–G6$^{AA}$ did not show detectable binding to $H_CE$ in this assay (Fig. 2c). These results suggest that most of the $H_CE$-interacting region is located in the middle to C-terminal portion of SV2A-L4.

To further validate this finding with full-length SV2 in neurons, we expressed SV2A containing either the wild-type SV2A-L4 or SV2Ac-L4 via lentiviral transduction in cortical neurons cultured from SV2A/B double knockout (KO) mice. These neurons mainly express SV2A and SV2B, but not SV2C[25,34]. We found that expression of SV2A and SV2Ac mediated similar levels of $H_CE$ binding to neurons (Fig. 2d). Furthermore, both SV2A and SV2Ac were able to mediate cell entry of BoNT/E and BoNT/A, resulting in cleavage of their neuronal substrate SNAP-25

(Fig. 2e and Supplementary Fig. 2). Taken together, these results demonstrate that SV2Ac maintains an SV2A-like binding capacity to BoNT/E on neurons.

## The structure of $H_CE$ in complex with SV2A

After prolonged efforts to rationally design and optimize a unique molecule that mimics the dual receptors of BoNT/E, we successfully determined the crystal structure of $H_CE$ in complex with SV2Ac–G6 at 2.59 Å resolution (Supplementary Table 1). There are two pairs of identical $H_CE$–SV2Ac–G6 complexes in one asymmetric unit, with each $H_CE$ bound with one molecule of SV2Ac and one G6 (Fig. 3a, b). The peptide linker between SV2Ac and G6 has no visible electron density, indicating a highly flexible conformation. G6, in the context of SV2Ac–G6 fusion protein, binds $H_CE$ in the same manner as the stand-alone G6, which further demonstrates that the peptide linker did not constrain SV2Ac and G6 association with $H_CE$ (Supplementary Fig. 3a).

The structure of SV2Ac-bound $H_CE$ is virtually identical to that in the context of BoNT/E holotoxin (PDB: 3FFZ, r.m.s.d. ~0.4 Å over 357 aligned Cα pairs)[50], suggesting the SV2-binding interface is largely pre-organized on BoNT/E. SV2Ac adopts a right-handed, quadrilateral β-helix fold, which is highly similar to SV2C-L4 observed in the $H_CA$–SV2C complex with a r.m.s.d. of ~0.3 Å between comparable Cα atoms (PDB: 5JLV)[13]. However, $H_CE$ binds to the side of the β-helical bundle of SV2Ac, which is in contrast to $H_CA$ which recognizes the open edge of the C-terminal β-strand of SV2C-L4 (Supplementary

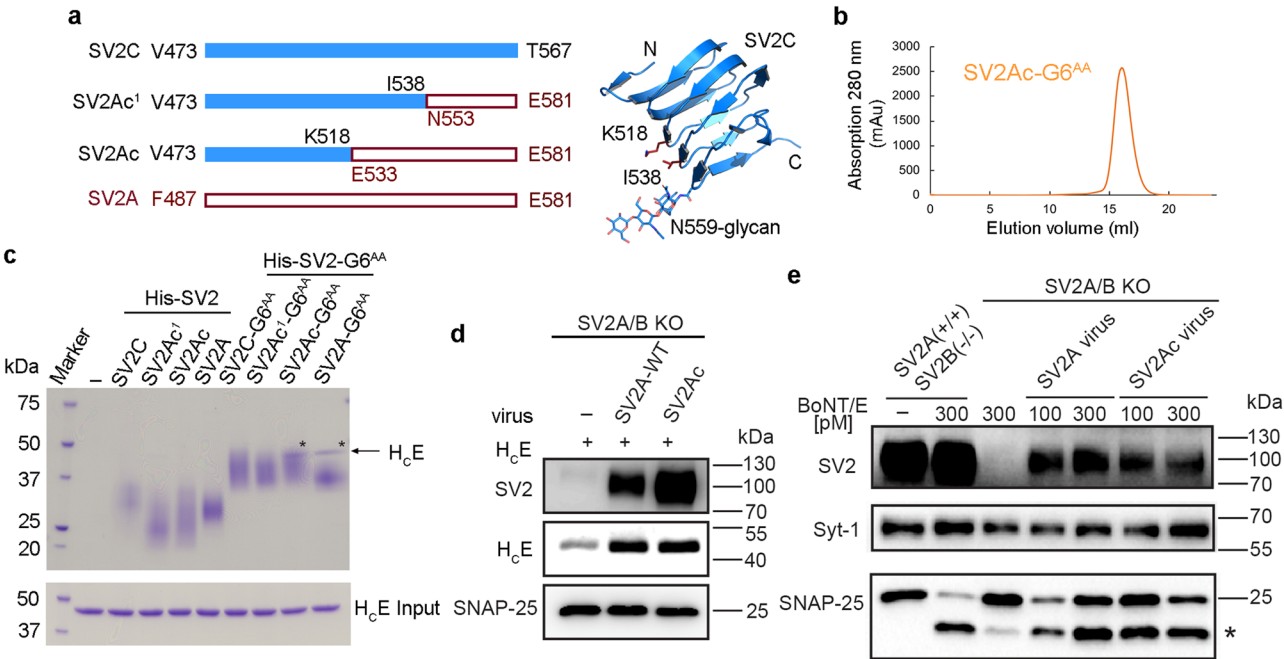

**Fig. 2 | An engineered SV2A–SV2C chimera maintains an SV2A-like binding to BoNT/E. a** A schematic diagram showing the designs of SV2A–SV2C chimeras where the sequences of SV2C-L4 and SV2A-L4 are shown as blue and red hollow bars, respectively. The structure of SV2C-L4 is shown as a blue cartoon with residues K518, I538, N559, and the N559 glycan shown as sticks (PDB: 5JLV). **b** The chimeric SV2Ac–G6$^{AA}$ fusion protein is mono-dispersed in solution based on a gel-filtration analysis. **c** A pull-down assay was performed using H$_C$E as prey and the His-tagged SV2-L4 or SV2–G6$^{AA}$ fusion proteins as baits. Only SV2A–G6$^{AA}$ and SV2Ac–G6$^{AA}$ were able to pull down H$_C$E (indicated by asterisks), but not other SV2–G6$^{AA}$ fusion proteins or the stand-alone SV2-L4 variants. A representative

result is shown ($n = 3$). **d** and **e** The full-length SV2A and SV2Ac were expressed via lentiviral transduction in mouse cortical neurons cultured from SV2A/B double knockout (KO) mice. Neurons were exposed to H$_C$E (5 min) (**d**) or BoNT/E at the indicated concentration (24 h in medium) (**e**). Cell lysates were harvested and analyzed by immunoblot assays. The synaptic vesicle protein, Synaptotagmin 1 (Syt-1), was detected as a loading control. The SNAP-25 antibody can detect both the full-length SNAP-25 and the fragment generated after cleavage by BoNT/E (marked with *). SV2A(+/+)/SV2B(−/−) neurons were analyzed in parallel as a positive control. A representative result is shown ($n = 2$). Source data are provided as a Source Data file.

Fig. 1c)[13,14]. This binding mode is consistent with our biochemical data showing that replacing the N-terminal region of SV2A with SV2C-like residues did not affect H$_C$E binding (Fig. 2c).

The SV2Ac-binding interface on H$_C$E is ~544 Å², which is completely located on H$_{CC}$E. It is composed of a central core interface involving extensive hydrogen bonds and hydrophobic interactions that are mediated by residues A1154, T1157, H1158, L1159, and F1160 of H$_C$E and residues N513, G514, R515, I517, E533, and Y535 of SV2Ac (Fig. 3c), and a separated interface where H$_C$E residues R1100 and K1102 establish hydrogen bonds with SV2Ac residues E537, Y557, N558, H578, and N579 (Fig. 3d and Supplementary Fig. 3c and Table 2). Consistent with the structural findings, mutating H$_C$E residues associated with this interface, such as H$_C$E$^{R1100G}$, H$_C$E$^{R1100G/K1102G}$, H$_C$E$^{H1158G}$, H$_C$E$^{T1157A/H1158G}$, and H$_C$E$^{A1154G/F1160G}$ abrogated binding to SV2Ac-G6$^{AA}$ in pull-down assays, which will be further discussed in a later section (Supplementary Fig. 3e). We also noticed a second interface between H$_C$E and SV2Ac that we attributed to a non-physiological crystal packing effect based on our observation that mutating key H$_C$E residues at this interface, such as K1173, N1207, and N1208, did not affect its interaction with SV2Ac-G6$^{AA}$ in pull-down assays (Supplementary Fig. 3b, d, e). On SV2Ac, all H$_C$E-interacting residues are located on one side of the β-helical bundle, and they are all native SV2A residues except for two amino acids located on the SV2C part of the chimera. Specifically, residue G514$^{SV2Ac}$ (equivalent to D514$^{SV2A}$ and G500$^{SV2C}$) forms a main-chain-mediated hydrogen bond with H$_C$E-H1158; R515$^{SV2Ac}$ (equivalent to R501$^{SV2C}$), which forms a hydrogen bond with H$_C$E-L1159, has a homologous substitution K515 on SV2A (Fig. 3c). When we replaced D514$^{SV2A}$ with an SV2C-like Gly, D514G$^{SV2A}$ could still maintain WT-like binding to H$_C$E (Supplementary Fig. 3f), and structural modeling showed that a Lys at

R515$^{SV2Ac}$ would not affect H$_C$E binding. Therefore, SV2Ac mimics the WT SV2A when recognizing H$_C$E.

## BoNT/E grips the sialic acid of SV2A glycan
In prior studies, we demonstrated that an N-glycan that is highly conserved on SV2A (N573), SV2B (N516), and SV2C (N559) across vertebrates is crucial for cell entry of BoNT/A and BoNT/E[13,25,51]. In our structure of the H$_C$E–SV2Ac–G6 complex, we only observed the electron density for the core N-acetylglucosamine (NAG) of this crucial glycan linked to SV2A-N573 (Supplementary Fig. 4a). Notably, this NAG is pointing away from the protein-protein interface between H$_C$E and SV2A (Fig. 4a), and given this binding mode, the rest of SV2A-N573 glycan core is unlikely able to interact with the neighboring H$_C$E residues. This is in sharp contrast to the glycan-binding mode of H$_C$A in which the quadruple-saccharide core of the N-glycan attached to SV2C-N559 is located next to the protein–protein interface where it can be conveniently gripped by H$_C$A via extensive interactions to enhance protein-based H$_C$A–SV2C binding (Supplementary Fig. 1c)[13]. These findings suggest that the SV2A-N573 glycan may adopt an unconventional H$_C$E-binding mode that is technically challenging to be defined by co-crystallization. This study was further complicated by the appearance of SV2Ac-G6 as smeared bands on SDS–PAGE gels representing heterogeneous glycoforms (Fig. 2c). This was not unexpected as recombinant glycoproteins expressed in HEK293 cells typically contain heterogeneous glycosylation under over-expression conditions[13,52].

During late-stage structure refinement, we noticed electron densities for an unknown molecule located in a pocket formed by four tyrosine residues (e.g. Y879, Y881, Y891, and Y1041) at the N-terminal sub-domain of H$_C$E (H$_{CN}$E), which is about ~25 Å away from N573 of

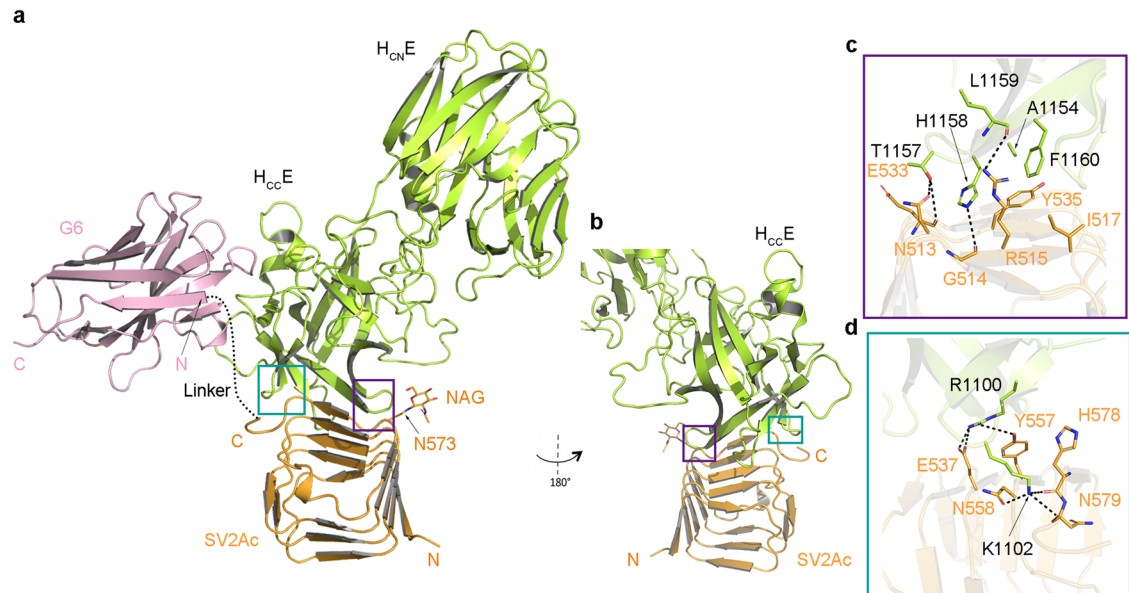

**Fig. 3 | Overall structure of the H$_C$E–SV2Ac–G6 complex. a** Cartoon representation of the H$_C$E–SV2Ac–G6 complex (H$_C$E, lemon; SV2Ac, orange; G6, pink). H$_C$E engages the protein moiety of SV2Ac mainly through two loops on its H$_{CC}$E domain (teal and purple boxed areas). The dotted line indicates the flexible linker between SV2Ac and G6 that was invisible in this structure. **b** The H$_C$E–SV2Ac interface is viewed from a different angle. **c** and **d** Close-up views of the interfaces between H$_C$E and SV2Ac that are highlighted in purple and teal boxes in (**a, b**) with key interacting residues shown as sticks.

SV2A (Supplementary Fig. 4b). Coincidentally, our earlier studies found that mutating residues Y879 or Y1041 on BoNT/E to alanine led to a dramatic 99% reduction of its neurotoxicity[26]. That observation could not be readily explained by any known mechanism, because all the known protein receptor- and ganglioside-binding sites are located on H$_{CC}$E[9]. Based on structural modeling studies, we found that a complex type of N-glycan attached to N573 of SV2A given the structure of the H$_C$E–SV2Ac complex could reach this distant pocket on H$_{CN}$E (Supplementary Fig. 4c)[53,54]. Moreover, we noticed that, besides the four Tyr residues, there are several other hydrophobic H$_C$E residues (e.g., Y926 and H1247) in this area that are also well-suited for carbohydrate binding. Taken together, we hypothesized that this unknown density could represent the distant portion of the SV2A-N573 glycan chain, which had a low occupancy at this remote site partly due to glycan heterogeneity.

The obscure electron density observed for this putative glycan indicated weak interactions that would be impractical to be characterized by direct binding studies. We, therefore, carried out systematic crystal soaking screens using component sugars of a typical complex type N-glycan, including monosaccharides sialic acid (Neu5Ac), N-acetylglucosamines (GlcNAc), galactose (Gal), and a disaccharide N-acetyl-D-lactosamine (Galβ1-4GlcNAc, LacNAc) that is the smallest repeating unit in most N-glycans. Based on a 2.77 Å resolution structure of a sialic acid-soaked H$_C$E–SV2Ac–G6 crystal, we could clearly see a sialic acid occupying this mysterious pocket on H$_{CN}$E (Fig. 4a and Supplementary Table 1). The electron densities at this site for all other sugars that were carried out in parallel crystal soaking studies were similar to the un-soaked crystals and could not be modeled. Structurally, this sialic acid is sandwiched between Y879 and Y1041 on H$_{CN}$E, surrounded by Y881, Y891, and H1247, and with associations further strengthened by several hydrogen bonds with R922 and N988 on H$_{CN}$E and G1248 of H$_{CC}$E (Fig. 4b). These BoNT/E residues are discontinued in the primary sequence, but converge in 3D to form a pocket that accommodates a sialic acid that is frequently found to cap the termini of oligosaccharide chains of N-glycans[53,55]. These structural findings suggest that BoNT/E appears to grip the terminal sialic acid of the SV2A-N573 glycan at a site that is distant (~25 Å away from N573 of SV2A) from the main protein-protein interface.

## BoNT/E and BoNT/A exploit distinct glycan-binding modes

Structural comparison between the glycan-bound H$_C$E and H$_C$A revealed that the sialic acid-binding site on H$_C$E is located close to the glycan-binding site on H$_C$A (Fig. 4c)[13]. We found that the glycan-binding residues on H$_C$A are not preserved on H$_C$E. For example, H$_C$A-G1292 is substituted by H$_C$E-Q1250 whose large side chain would clash with the SV2C glycan (Fig. 4d). This may preclude H$_C$E from using an H$_C$A-like glycan-binding mode. On the other hand, the sialic acid-binding site on H$_C$E is partially conserved on H$_C$A (Fig. 4e), raising the possibility that BoNT/A might use this H$_C$E-like site to recognize the terminal sialic acid of the SV2C glycan. However, our structural modeling reveals that the glycan anchoring residue N559 of the H$_C$A-bound SV2C is located very close to this hypothetic H$_C$E-like sialic acid-binding site (Fig. 4c), and as a result, the terminal sialic acid of the H$_C$A-bound SV2C N559 glycan would be located beyond this hypothetic sialic acid-binding pocket on H$_C$A. In one of our earlier mutagenesis studies on BoNT/A, we found that mutating H$_C$A residues N905, F917, and D1289, which are equivalent to the sialic acid-binding residues Y879, Y891, and H1247 on H$_C$E, displayed only moderately decreased neurotoxicity[56]. These results suggest that H$_C$A does not use an H$_C$E-like glycan-binding mode, as the core saccharide of the SV2C-N559 glycan together with the protein moiety of SV2C plays a dominant role in mediating BoNT/A binding[13].

## Simultaneous binding to the protein- and glycan-moiety of SV2A is crucial for BoNT/E function

We then carried out structure-based mutagenesis studies to validate the structural findings and to further characterize the functional role of BoNT/E–SV2A interplays. Guided by the crystal structures, we designed H$_C$E variants that carry two different types of mutations: (1) mutations that weaken H$_C$E binding to SV2A protein moiety, including H$_C$E$^{R1100G}$, H$_C$E$^{K1102G}$, H$_C$E$^{R1100G/K1102G}$, H$_C$E$^{T1157A}$, H$_C$E$^{H1158G}$, H$_C$E$^{T1157A/H1158G}$, and H$_C$E$^{A1154G/F1160G}$, and (2) mutations that disrupt H$_C$E association with sialic acid, including H$_C$E$^{Y879G/Y881G}$, H$_C$E$^{Y891G/Y1041G}$, and H$_C$E$^{E1246A/H1247A}$. We first confirmed that all these mutations did not alter H$_C$E folding and stability based on thermal denaturation experiments (Supplementary Fig. 5). We then carried out two sets of studies to examine how these H$_C$E mutants recognized SV2A-G6$^{AA}$ in vitro using pull-down assays

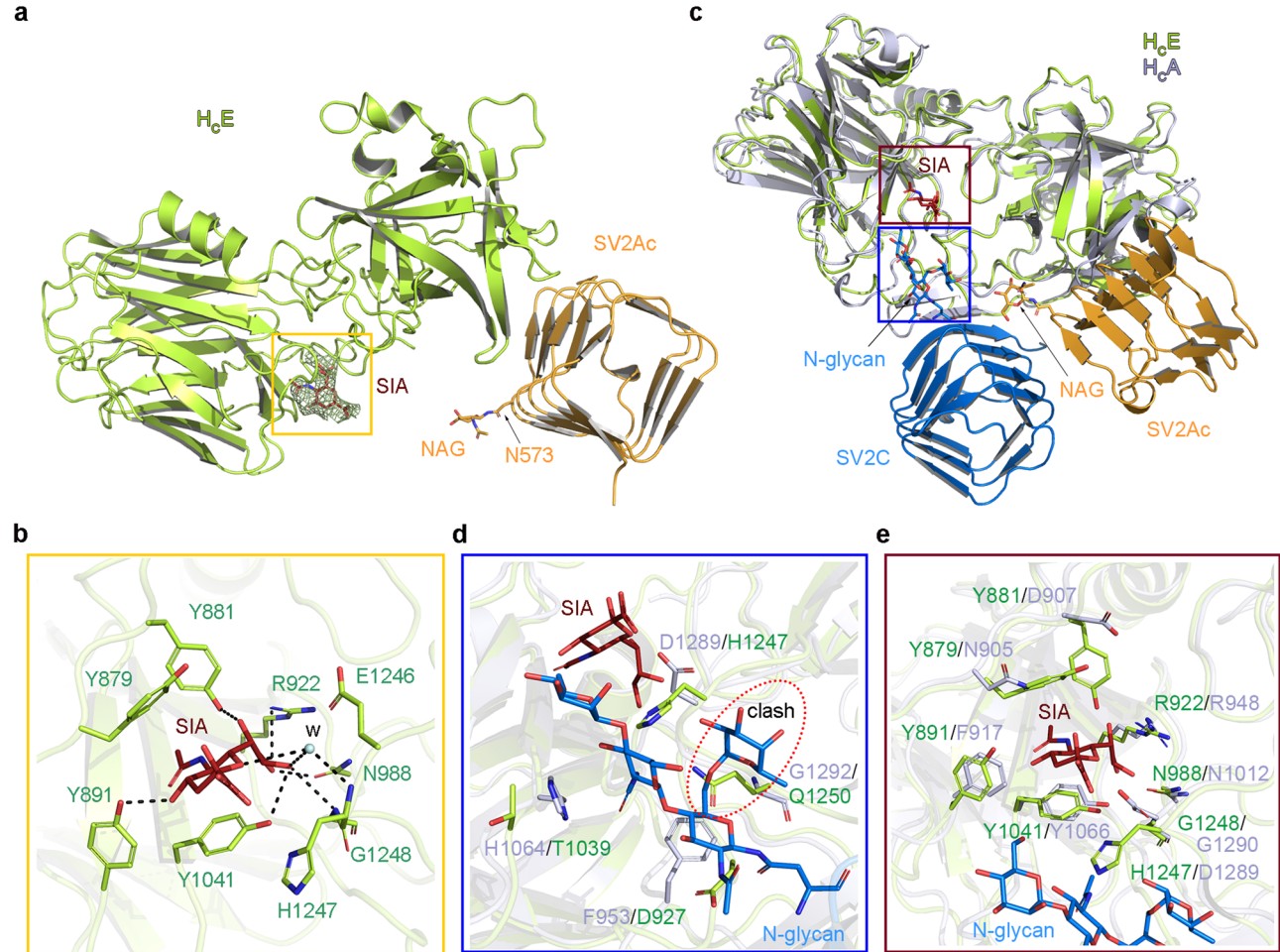

**Fig. 4 | Structure of the sialic acid-bound $H_CE$–SV2Ac complex and comparison of the glycan-binding modes between BoNT/E and BoNT/A. a** Cartoon representation of the sialic acid (SIA)-bound $H_CE$–SV2Ac complex with $H_CE$ colored in lemon, SV2Ac in orange, and sialic acid in red. An omit electron density map for sialic acid contoured at 1.5$\sigma$ was overlaid with the final refined model. **b** A close-up view of the interactions between $H_CE$ and sialic acid. One water molecule (W) that mediates the $H_CE$–SIA binding is shown as a cyan sphere. **c** The $H_CE$–SV2Ac–SIA complex and the $H_CA$–SV2C complex (PDB: 5JLV) were superimposed based on $H_CE$ (lemon) and $H_CA$ (blue-white). The SIA (red), NAG (orange), and SV2C glycan (blue) are shown as sticks. The SIA-binding and the SV2C-glycan-binding pockets are highlighted in red and blue boxes, respectively. **d** A close-up view into the SV2C-glycan-binding pocket reveals the differences between $H_CA$ (blue-white) and $H_CE$ (lemon) in this area. $H_CE$ residue Q1250 would clash with the SV2C glycan based on the superposition. **e** A close-up view into the SIA-binding pocket shows that this pocket is partly conserved between $H_CE$ and $H_CA$.

(Supplementary Fig. 3e) and endogenous SV2A and SV2B in cultured rat cortical neuron (Fig. 5a). We found that $H_CE^{R1100G}$, $H_CE^{R1100G/K1102G}$, $H_CE^{H1158G}$, $H_CE^{T1157A/H1158G}$, and $H_CE^{A1154G/F1160G}$ that have the disrupted protein-based SV2A-binding interface showed largely abolished binding to SV2A-G6$^{AA}$ in vitro and endogenous SV2 on rat cortical neurons. Furthermore, mutating the sialic acid-binding residues in $H_CE^{Y879G/Y881G}$, $H_CE^{Y891G/Y1041G}$, and $H_CE^{E1246A/H1247A}$ also largely reduced their binding to SV2 on neurons (Fig. 5a). We did not observe detectable changes of binding between SV2A-G6$^{AA}$ and $H_CE^{Y879G/Y881G}$ or $H_CE^{Y891G/Y1041G}$ in vitro using pull-down assays (Supplementary Fig. 3e), which could be due to the heterogeneous glycosylation of the recombinant SV2A that is different from the glycosylation pattern of SV2A on neurons. Therefore, the in vitro pull-down assay was mostly detecting the protein-mediated interactions. Taken together, these data suggest that both the protein- and the glycan-mediated associations are necessary for $H_CE$–SV2A recognition on neuronal surfaces.

To further establish the physiological relevance of the protein- and glycan-mediated $H_CE$–SV2A interactions, we produced four SV2A-binding deficient mutants of the full-length BoNT/E based on the results of the mutagenesis studies on $H_CE$ described above, and examined their neurotoxicity at motor nerve terminals using an ex vivo

mouse phrenic nerve hemi-diaphragm (MPN) assay (Fig. 5b and Supplementary Fig. 6)[57]. We found that BoNT/E$^{R1100G/K1102G}$ and BoNT/E$^{H1158G/F1160G}$, which have mutations at two separated sites of the protein–protein interface with SV2A, showed ~90% decreased neurotoxicity, while BoNT/E$^{R1100G/H1158G/F1160G}$ that carries mutations at both sites showed a further decrease to ~99% (Fig. 5b). To examine the functional role of glycan-mediated interactions, we designed BoNT/E$^{Y879G/Y1041G}$ based on $H_CE^{Y879G/Y881G}$ and $H_CE^{Y891G/Y1041G}$, in which both Y879 and Y1041 of BoNT/E that sandwich the sialic acid were mutated. Remarkably, BoNT/E$^{Y879G/Y1041G}$ only retained ~0.1% neurotoxicity despite its intact binding site for the protein moiety of SV2A, strongly supporting the direct involvement of the N-glycan of SV2A and SV2B in BoNT/E binding and function[25]. The destructive effects of mutations at the glycan-binding site of BoNT/E were stronger as revealed by the MPN assay in comparison to the results of neuron binding assay based on $H_CE$ (Fig. 5a), which is likely due to the different functional read-out sensitivity of the two assays and the different amount of BoNT/E (WT at 2–8 pM and mutants at 20 pM–6 nM) and $H_CE$ (200 nM) used. Together, these data demonstrate that both the protein and glycan moieties of SV2A are essential for the neurotoxicity of BoNT/E at motor nerve terminals.

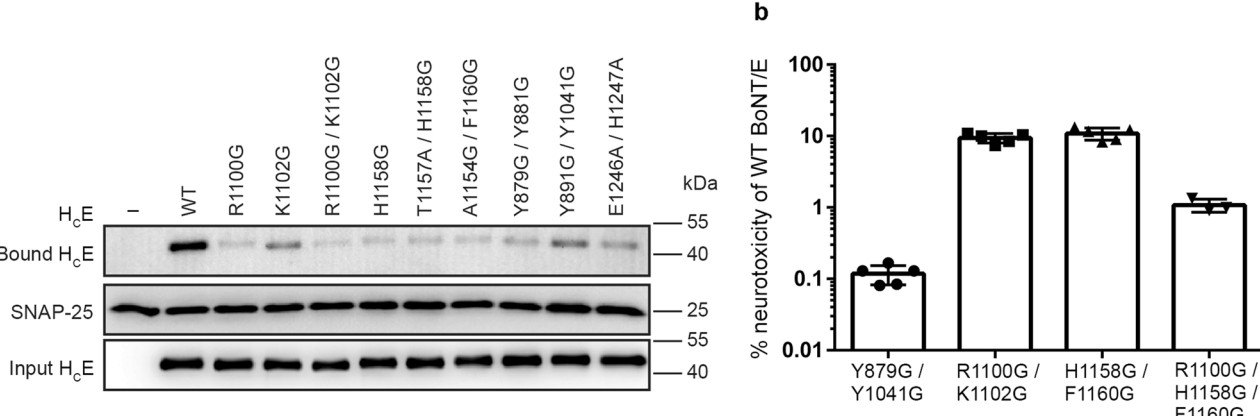

**Fig. 5 | Simultaneous binding to the protein- and glycan-moiety of SV2A is crucial for neuronal binding and neurotoxicity of BoNT/E. a** Rat cortical neurons were exposed to a high K⁺ buffer containing 200 nM H$_C$E for 5 min at 37 °C. Cells were washed three times and binding of the biotinylated H$_C$E variants was detected by immunoblot analysis of cell lysates using Streptavidin-HRP. SNAP-25 was detected as a loading control. **b** MPN assay showed drastically decreased neurotoxicity of BoNT/E when its protein–protein and protein–sialic acid-binding interfaces, respectively, were disrupted by mutagenesis. Graph shows means ± s.d. of *n* = 3 biologically independent experiments for triple mutant R1100G/H1158G/F1160G and *n* = 5 for all other mutants. Source data are provided as a Source Data file.

## BoNT/E selectively recognizes SV2A and SV2B, but not SV2C

Our previous studies suggest that BoNT/E can utilize SV2A and SV2B, but not the closely related SV2C, as receptors in hippocampal and cortical neurons[25,26]. Since both SV2A/SV2B and SV2C have a conserved N-glycan at the same location (N573$^{SV2A}$, N516$^{SV2B}$, and N559$^{SV2C}$), we hypothesized that H$_C$E may distinguish SV2A/SV2B from SV2C mainly based on amino acid differences at the protein–protein interface. Structure-based sequence analyses revealed that the BoNT/E-binding residues are mostly identical between SV2A and SV2B except for three subtle amino acid substitutions, which are H578$^{SV2A}$ and N579$^{SV2A}$ that form main-chain-mediated hydrogen bonds with H$_C$E-K1102, and I517$^{SV2A}$ that packs against the hydrophobic H$_C$E-F1160. These three SV2A residues are replaced by E521$^{SV2B}$, Q522$^{SV2B}$, and T460$^{SV2B}$, respectively, which should not have a major effect on H$_C$E binding (Fig. 6a and Supplementary Table 2). In contrast, Y535$^{SV2A}$/Y478$^{SV2B}$ located at the core of the BoNT/E–SV2 protein–protein interface is replaced with T521$^{SV2C}$, which will weaken the hydrophobic packing with H$_C$E-F1160, and the interaction between Y557$^{SV2A}$/Y500$^{SV2B}$ and H$_C$E-R1100 will be disrupted by the corresponding residue D543$^{SV2C}$ in human/mouse (homologous E543$^{SV2C}$ on the rat) (Fig. 6a and Supplementary Table 2).

To test these predictions, we swapped residues Y535 and Y557 on SV2A-L4 with the corresponding residues on SV2C-L4 to generate an "SV2C-like" SV2A$^{Y535T/Y557D}$ and vice versa to generate an "SV2A-like" SV2C$^{T521Y/D543Y}$. We first linked them with G6$^{AA}$ and examined how they recognized H$_C$E in vitro using a bio-layer interferometry (BLI) assay. We found that SV2A$^{Y535T/Y557D}$–G6$^{AA}$ showed a markedly decreased binding to H$_C$E vs. SV2A–G6$^{AA}$, while SV2C$^{T521Y/D543Y}$–G6$^{AA}$ showed a clearly improved binding to H$_C$E vs. SV2C–G6$^{AA}$ (Supplementary Fig. 7). To better understand the physiological relevance of this structural finding, we expressed these two mutants as full-length SV2 (SV2A$^{Y535T/Y557E}$ and SV2C$^{T521Y/E543Y}$, the rat SV2 genes that has E543 on SV2C were used in this experiment) in cortical neurons cultured from SV2A/SV2B KO mice via lentiviral transduction. Using the wild-type BoNT/E, we found that the SV2A$^{Y535T/Y557E}$ mutant lost its function to mediate toxin entry at three different toxin doses tested (Fig. 6b). Expression of SV2C$^{T521Y/E543Y}$ mediated a low level of entry of BoNT/E, resulting in a minor cleavage of SNAP-25 at two toxin doses tested, whereas over-expression of WT SV2C did not mediate entry of BoNT/E (Fig. 6c). Both SV2C$^{T521Y/E543Y}$ and WT SV2C mediated entry of BoNT/A (Supplementary Fig. 8). These results suggest that SV2C$^{T521Y/E543Y}$ gained the capability to mediate BoNT/E entry, albeit at a low efficacy. Additional mutations might be needed to further enhance the binding of BoNT/E to SV2C$^{T521Y/E543Y}$.

These findings suggest that BoNT/E is able to detect the subtle differences in the primary sequences of SV2A/2B and SV2C, even though the overall structures of SV2A and SV2C are similar. In contrast, BoNT/A recognizes all three SV2 isoforms because there are mostly backbone-to-backbone interactions between BoNT/A and SV2 at the protein-protein interface that tolerate residue changes across SV2 isoforms[13,14].

## Discussion

BoNT/E together with BoNT/A and BoNT/B are the major causes of human botulism. Paradoxically BoNT/A and BoNT/B are also approved drugs for a myriad of therapeutic and esthetic uses. Due to its unique pharmacological and clinical profiles, BoNT/E has attracted growing therapeutic interests and is currently in clinical trials for new indications that may benefit from BoNT/E's faster onset of action and shorter duration[19,20,58,59]. Here, we determined the crystal structure of H$_C$E in complex with a fusion protein of human SV2A and SV2C, which reveals two distant receptor-binding sites that are well separated on the two subdomains of H$_C$E: the major interface is between H$_{CC}$E and the side of the quadrilateral β-helix of SV2A-L4, while the SV2A-N573 glycan extends toward H$_{CN}$E with one of its terminal sialic acids buried in a hydrophobic pocket on H$_{CN}$E (Fig. 7a). This is distinct from BoNT/A, which uses a composite interface located between H$_{CN}$A and H$_{CC}$A to recognize both the protein component of SV2C and the core saccharides of a neighboring N-glycan (Fig. 7a and Supplementary Fig. 1c)[13]. Further structure-based mutagenesis and functional studies demonstrate that both the protein- and N-glycan-based engagements are crucial for SV2A-mediated binding and entry of BoNT/E into neurons (Fig. 5a, b and Supplementary Fig. 3e).

The unexpected recognition of the terminal sialic acid of a conserved N-glycan on SV2A by BoNT/E is in fact reminiscent of the well-studied receptor binding strategy of the influenza virus, whose HA proteins exploit the terminal sialic acids on host glycoproteins and glycolipids as cellular receptors[60]. A similar strategy is also used by HA70, a nontoxic component of the large 14-subunit progenitor toxin complex (L-PTC) of BoNT/A, to recognize sialic acids as its carbohydrate receptors on the intestinal epithelial cell surface for enrichment and absorption[61,62]. It is tempting to speculate that BoNT/E may use this glycan-binding site to recognize sialic acids on neuronal glycoproteins and/or glycolipids before encountering SV2. In contrast to viruses and other toxins that typically use multivalent-binding modes to compensate for the weak association at each individual protein–glycan

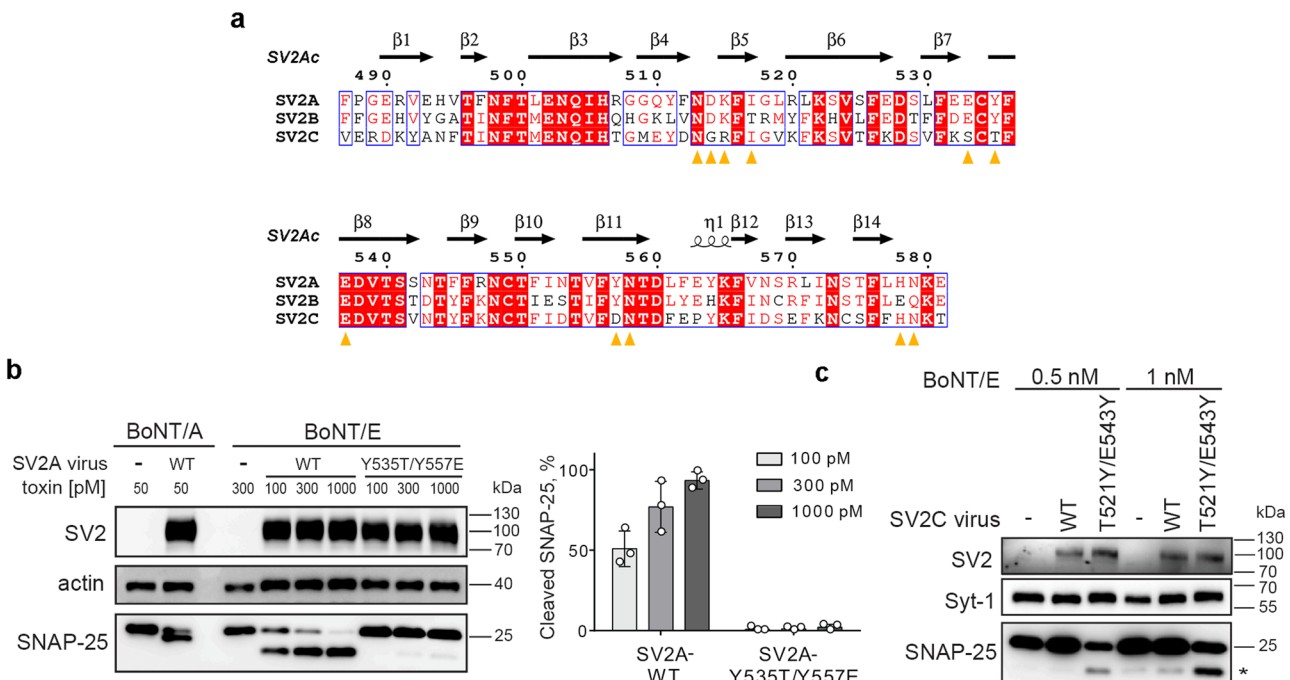

**Fig. 6 | BoNT/E differentiates SV2A and SV2B from SV2C. a** Amino acid sequence alignment among human SV2A, SV2B and SV2C in the L4 region (prepared using MultAlin[75] and ESPript 3.0[76]). Identical residues are indicated with white letters on a red background, conserved residues are in red letters, and varied residues are in black letters. The SV2A residues that are recognized by $H_CE$ are indicated by orange triangles. Residue numbers of SV2A and the secondary structures of chimeric SV2Ac are shown on the top. **b** The WT SV2A or SV2A^Y535T/Y557E mutant was expressed in SV2A/B KO mouse cortical neurons via lentiviral transduction. Neurons were exposed to the indicated toxins (14 h in medium). Cell lysates were harvested and analyzed by immunoblot assays. Actin served as a loading control. Left panel: representative immunoblots. Right panel: the percentage of SNAP-25 cleavage by BoNT/E in the left panel was quantified using ImageJ by comparing the amount of cleavage products versus the intact SNAP-25. Error bar represents SD from three independent experiments. **c** The WT SV2C or SV2C^T521Y/E543Y mutant was expressed in SV2A/B KO neurons. Neurons were exposed to the indicated concentrations of BoNT/E (24 h in the medium). Cell lysates were harvested and analyzed by immunoblot assays. A representative result is shown (n = 2). The cleaved SNAP-25 is marked with *. Source data are provided as a Source Data file.

interface, our studies suggest that BoNT/E makes use of an independent protein–protein interface to not only enhance glycan-mediated binding to SV2 but also simultaneously provide the crucial specificity information to determine its tissue tropism (Fig. 7). We found that there are several amino acid substitutions at the protein-mediated SV2A-binding interface on $H_CE$ among 12 known BoNT/E subtypes (BoNT/E1–E12). For example, a key SV2A-binding residue R1100 is replaced with S1100 on subtype BoNT/E10 and E11, which could weaken receptor binding and may be partly responsible for the reported lower toxicity of culture supernatants containing BoNT/E10 and E11 besides other factors such as growth rate and toxin secretion[63] (Supplementary Fig. 9). However, the SV2 glycan-binding sites are highly conserved in all twelve BoNT/E subtypes (Supplementary Fig.9), which should allow certain tolerance for amino acid changes at the SV2 protein-binding site of BoNT/E during evolution. Furthermore, this glycan-binding site on BoNT/E is partially preserved on BoNT/A, which is very close to the known glycan-binding site on $H_CA$ that accommodates the core saccharides of the SV2C N559 glycan[13], suggesting possible additional interactions between BoNT/A and this SV2C glycan.

It is well accepted that the docking orientations of BoNTs, with each toxin composed of a light chain (LC), the translocation domain ($H_N$), and the receptor-binding $H_C$, on the neuronal surface are largely constrained by simultaneous binding of the $H_C$ to the membrane-anchored gangliosides and protein receptors[7–10]. We found that, even though SV2 binds to distinct sites on $H_CE$ and $H_CA$ and uses different glycan-binding modes, the putative docking orientations of $H_CE$ and $H_CA$ on the cell surface are similar and the relative orientations of the quadrilateral β-helix of SV2A-LC and SV2C-L4 are also similar (Fig. 7a, b). However, in the context of the holotoxins, the LC-$H_N$ moiety of BoNT/E and BoNT/A are orientated

differently relative to the $H_C$ and the membrane. This is because BoNT/A displays a linear "open-wing"-like arrangement where the $H_C$ and LC are located on opposite sides of the long helical $H_N$, while the $H_C$ and the LC-$H_N$ of BoNT/E fold toward each other resulting in a "closed-wing" conformation (Fig. 7c, d)[50,64]. This finding provides the structural basis to inform future studies on how BoNTs may reorganize their three domains after receptor-mediated binding on neuron surface and proceed to transmembrane delivery of the LC to the cytosol, as prior studies suggested that the translocation process is more rapid in BoNT/E than BoNT/A[18,23].

SV2A is expressed in a subset of motor neurons, whereas both SV2B and SV2C are detected in the majority of motor neurons[31–33]. In prior studies, we demonstrated that BoNT/E cannot utilize SV2C as a receptor in cultured hippocampal and cortical neurons[25]. Whether SV2C in motor neurons may still function as a receptor for BoNT/E remains to be determined[31,32]. Nevertheless, our studies provide a structural basis to understand the differences in BoNT/E recognition of SV2A/2B versus SV2C, particularly involving their protein sequences. The structures suggest a potentially important role of the conserved SV2 N-glycan in mediating BoNT/E interactions, which may also contribute to differences between SV2A/2B and SV2C as BoNT/E receptors.

As the three SV2 isoforms have different tissue distributions in human[34,35], the variations in binding affinity toward SV2 isoforms between BoNT/E and BoNT/A may contribute to their distinct pharmacological and therapeutic features[19,20,58,59]. This knowledge could be harnessed to engineer new BoNT/E variants with modified specificities toward different SV2 isoforms or help to fine-tune BoNT/A–SV2 interplays for new clinical developments. Notably, recent structure-based engineering of BoNT/B successfully enhanced its binding to human receptor synaptotagmin-II and led to improved clinical

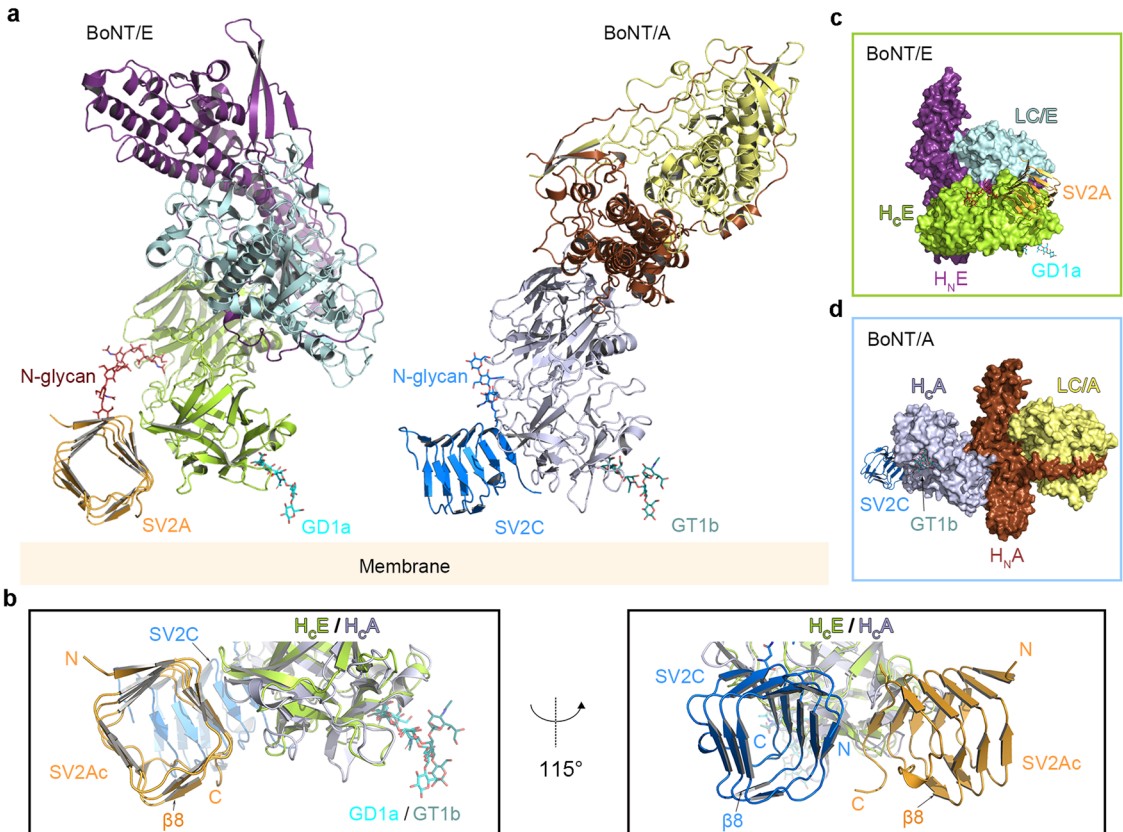

**Fig. 7 | Proposed models for simultaneous binding of BoNT/E and BoNT/A to the membrane-anchored SV2 and gangliosides. a** Proposed binding modes of BoNT/E (PDB: 3FFZ) and BoNT/A (PDB: 3BTA) with the membrane-bound SV2 and gangliosides. The holotoxins are positioned based on the structures of the $H_CE$–SV2Ac and the $H_CA$–SV2C (PDB: 5JLV) complexes. A representative complex-type N-glycan (red sticks, PDB: 3QUM) is modeled to represent the N-glycan of SV2A. The gangliosides GD1a and GT1b are modeled based on the structures of $H_CE$–GD1a (PDB: 7OVW) and $H_CA$–GT1b (PDB: 2VU9). **b** The structures of the $H_CE$–SV2Ac and the $H_CA$–SV2C (PDB: 5JLV) complexes with the modeled gangliosides are superimposed based on $H_CE$ and $H_CA$. The view angle on the left panel is

identical to that shown in panel (**a**). A different view with a rotation of -115° about a vertical axis is shown on the right panel. Two equivalent β-sheets (β8) on SV2Ac-L4 and SV2C-L4 are highlighted as a marker to show the similar orientations of SV2Ac and SV2C relative to the membrane in this putative model. **c** Surface representation of BoNT/E holotoxin in complex with SV2A (orange) and GD1a (cyan sticks). BoNT/E adopts a "closed-wing" conformation, in which $H_CE$ (lemon) and LC/E (pale cyan) are located on the same side of the long helical $H_NE$ (purple). **d** Surface representation of BoNT/A holotoxin in complex with SV2C (blue) and GT1b (deep teal sticks). BoNT/A has an "open-wing" conformation, in which $H_CA$ (light purple) and LC/A (yellow) are located on the opposite sides of the long helical $H_NA$ (brown).

efficacy[15,16,65,66]. At the same time, the highly conserved glycan-binding pocket among all BoNT/E subtypes is of particular interest for the future development of epitope-focused antibodies for the counter-measure of botulism or reversal of muscle paralysis in the clinic.

## Methods

### Ethics statement

All animal studies in the Dong lab were approved by the Boston Children's Hospital Institutional Animal Care and Use Committee (Protocol Number: 18-10-3794R). All procedures were approved by the Institute of Biosafety Committees at Boston Children's Hospital (Protocol Number: IBC-P00000501). The MPN assay (project license 2018/209) was performed in the Rummel lab according to §4 Abs. 3 (killing of animals for scientific purposes, German animal protection law (TSchG)). The numbers of animals sacrificed by trained personnel before the dissection of organs were reported yearly to the animal welfare officer of the Central Animal Laboratory and to the local authority, Veterinäramt Hannover.

### Cloning, expression, and purification of recombinant proteins

The genes encoding $H_CE$ (residues R846–K1252) and VHH G6 (residues Q1–S129) were cloned into a modified pET28a vector with a 6xHis/SUMO (*Saccharomyces cerevisiae* Smt3p) tag introduced to the N-terminus. The core regions of human SV2A-L4 (residues F487–E581)

and human SV2C-L4 (residues V473–T567) were cloned into a modified pcDNA vector for mammalian cell expression, and a human IL2 signal sequence (MYRMQLLSCIALSLALVTNS), a 9xHis tag, a factor Xa-cleavage site, and a human rhinovirus 3C protease cleavage site were added to the N-terminus. The chimeric SV2A–SV2C-L4 constructs were generated by two-step PCR and verified by DNA sequencing. Specifically, SV2Ac[I] was made by replacing SV2A amino acids F487–I552 with the corresponding SV2C amino acids V473–I538 and SV2Ac was made by replacing SV2A residues F487–E532 with the corresponding SV2C residues V473–K518. The chimeras were cloned into the modified pcDNA vector for expression. For the SV2Ac-G6 fusion protein, G6 or the G6[AA] (D100A/D115A) mutant was covalently linked to the C terminus of SV2Ac through a 10-amino acid peptide linker (GTSPSASGGS) and cloned into the modified pcDNA vector for expression. The other fusion constructs, including SV2A–G6[AA], SV2Ac[I]–G6[AA], and SV2C–G6[AA], were generated in a similar manner. All site-specific mutations were generated by two-step PCR and verified by DNA sequencing.

$H_CE$ and VHH G6 (WT and mutations) were expressed in *E. coli* strain BL21-Star (DE3) (Invitrogen). Bacteria were cultured at 37 °C in an LB medium containing kanamycin. The temperature was reduced to 18 °C when $OD_{600}$ reached ~0.8. Expression was induced with 1 mM IPTG (isopropyl-b-D-thiogalactopyranoside) and continued at 18 °C overnight. The cells were harvested by centrifugation and stored at −80 °C until use.

The 6xHis/SUMO-tagged $H_CE$ and G6 (WT and mutations) were purified using $Ni^{2+}$-NTA (nitrilotriacetic acid, Qiagen) affinity resins in a buffer containing 50 mM Tris, pH 7.5, 400 mM NaCl, and 40 mM imidazole. The proteins were eluted with a high-imidazole buffer (50 mM Tris, pH 7.5, 400 mM NaCl, and 300 mM imidazole) and then exchanged into a buffer containing 50 mM Tris, pH 7.5, and 400 mM NaCl. The 6xHis/SUMO tags of $H_CE$ and G6 were cleaved by SUMO protease. $H_CE$ was further purified by MonoS ion-exchange chromatography (GE Healthcare) in a buffer containing 20 mM MES, pH 6.0, 150 mM NaCl, and 1 mM TCEP, and eluted with a NaCl gradient. The peak fractions of $H_CE$ were pooled and subjected to $Ni^{2+}$-NTA re-binding, and the flow through was concentrated, frozen in liquid nitrogen, and kept at −80 °C. G6 was also further purified by $Ni^{2+}$-NTA re-binding. To obtain the $H_CE$–G6 complex for crystallization, the purified $H_CE$ was mixed with G6 for 1-h incubation, then purified by Superdex-200 SEC (GE Healthcare) in a buffer containing 20 mM HEPES, pH 7.5, and 150 mM NaCl, and the peak fractions were concentrated to ~10 mg/ml for crystallization.

SV2A, SV2Ac[1], SV2Ac, SV2C, SV2A–G6[AA], SV2Ac[1]–G6[AA], SV2Ac–G6[AA], SV2Ac–G6, SV2C–G6[AA], and their mutations were expressed, and secreted from FreeStyle HEK 293 cells (ThermoFisher) and purified directly from cell culture media using $Ni^{2+}$-NTA resins. To prepare the $H_CE$–SV2Ac–G6 complex for crystallization, the purified $H_CE$ and 9xHis-tagged SV2Ac–G6 were mixed at a molar ratio of ~2:1 for 2 h at 12 °C. The complex was isolated using $Ni^{2+}$-NTA resins and further purified by Superdex-200 SEC (GE Healthcare) in a buffer containing 10 mM HEPES, pH 7.5, 150 mM NaCl, and 1 mM TCEP. The N-terminal 9xHis tag of SV2Ac–G6 in the complex was removed by 3C protease, and the complex was further purified by $Ni^{2+}$-NTA re-binding and concentrated to ~8 mg/ml for crystallization.

The wild-type and mutated recombinant full-length activated BoNT/E1 were produced under biosafety level 2 containment (project number GAA A/Z 40654/3/123/3) recombinantly in *E. coli* BL21 DE3 strain in Dr. Rummel's lab in Germany as described previously[67]. All mutations were generated by two-step PCR and verified by DNA sequencing. BoNT/E and mutants carrying C-terminal His6-tag were purified on $Co^{2+}$-Talon matrix (Takara Bio Europe S.A.S., France) and eluted with 50 mM Tris–HCl (pH 8.0), 150 mM NaCl, and 250 mM imidazole. For proteolytic activation and removal of an affinity tag, BoNT/E was incubated for 16 h at room temperature with 0.01 U bovine thrombin (Sigma-Aldrich Chemie GmbH, Germany) per μg of BoNT. Subsequent gel filtration (Superdex-200 SEC; GE Healthcare, Germany) was performed in phosphate-buffered saline (pH 7.4). For storage, BoNT/E and mutants were shock-frozen in liquid nitrogen and stored at −80 °C.

## Crystallization
Initial crystallization screens of the $H_CE$–G6 and the $H_CE$–SV2Ac–G6 complex were carried out at 18 °C using a Gryphon crystallization robot (Art Robbins Instruments) with high-throughput crystallization screening kits (Hampton Research and Qiagen). The original crystals of the $H_CE$–G6 complex were obtained in a reservoir containing 0.2 M NaCl and 20% PEG 3350. And the $H_CE$–SV2Ac–G6 complex was originally crystallized in a reservoir containing 0.2 M potassium sulfate and 20% PEG 3350. Extensive manual optimization was then performed using the hanging-drop vapor-diffusion method via mixing the protein with reservoir solution at a 1:1 ratio. For the $H_CE$–G6 complex, the best crystals were obtained in a reservoir containing 0.1 M HEPES, pH 7.0, 0.2 M NaCl, and 18% PEG 3350, and the crystals were cryoprotected in the mother liquor supplemented with 20% (v/v) ethylene glycol. The best crystals for the $H_CE$–SV2Ac–G6 complex were obtained in a reservoir containing 0.1 M HEPES, pH 7.5, 0.2 M potassium sulfate, 20% PEG 3350, and 5% PEG 400. Streak seeding was necessary to obtain single crystals. For the sugar soaking studies, the crystals of the $H_CE$–SV2Ac–G6 complex were soaked in the mother liquor supplemented with 100 mM sialic acid (Neu5Ac), N-acetylglucosamines (GlcNAc), galactose (Gal), or N-acetyl-D-lactosamine (Galβ1-4GlcNAc, LacNAc) at 18 °C overnight. The crystals were then cryoprotected in buffers containing 0.1 M HEPES, pH 7.5, 23% PEG3350, 12% glycerol, 0.16 M potassium sulfate, and the corresponding sugars, and flash-frozen in liquid nitrogen for diffraction studies.

## Data collection and structure determination
The X-ray diffraction data were collected at 100 K at the NE-CAT beamline 24-ID, Advanced Photon Source (APS). The data were processed with XDS as implemented in RAPD (https://github.com/RAPD/RAPD)[68]. The complex structures were solved by the molecular replacement software PHENIX.Phaser[69] using the structures of $H_CE$ (PDB: 3FFZ)[50], VHH (PDB: 6GLW)[70], and SV2C-L4 (PDB: 5JLV)[13] as the search models. The crystals of the $H_CE$–G6 complex belong to space group $P2_1 2_1 2_1$ and there are five pairs of the $H_CE$–G6 complexes in one asymmetric unit. The crystals of the $H_CE$–SV2Ac–G6 complex belong to space group $C2 2 2_1$ with two pairs of the $H_CE$–SV2Ac–G6 complexes in the asymmetric unit. The initial atomic models were refined with Phenix.Refinement[69]. Further structural modeling and refinement were carried out iteratively using COOT[71] and Phenix.Refinement[69] or Refmac5 refinement[72]. The structure of the sialic acid-bound $H_CE$–SV2Ac–G6 complex was solved using the $H_CE$–SV2Ac–G6 complex as a model and sialic acid was modeled based on the $F_O$–$F_C$ electron density maps. All the refinement progresses were monitored with the free $R$ value using a 5% randomly selected test set[73] and the structures were validated by MolProbity[74]. Data collection and structural refinement statistics are listed in Supplementary Table 1. All structure figures were prepared using Pymol (DeLano Scientific).

## Liposome co-sedimentation assay
Large unilamellar vesicles (LUV) were prepared as previously described[40]. Briefly, lipids (1,2-dioleoyl-sn-glycero-3-phospho-L-serine (DOPS) and 1-palmitoyl-2-(9,10-dibromostearoyl) phosphatidylcholine (BrPC)) (Avanti Polar Lipid) were dissolved in chloroform while GT1b trisodium salt (Santa Cruz Biotechnology) was dissolved in methanol. The lipids (70/20/10 mol% BrPC/DOPS/GT1b) were mixed, dried under nitrogen gas, and then placed under vacuum for overnight. The dried lipids were rehydrated and subjected to 5–10 rounds of freezing and thawing cycles. Liposomes were prepared by extrusion through a 200 nm pore membrane using an Avanti Mini Extruder according to the manufacturer's instructions.

The $H_CE$–GT1b binding experiment was conducted by mixing 1 μM of $H_CE$ or $H_CE$ pre-incubated with 2 μM of G6 with 200 μM of liposomes. The protein–liposome mixture was then incubated in a buffer containing 100 mM NaCl and 20 mM HEPES (pH 7.0) at room temperature for 1 h followed by spinning progressively at 4000×g, 9000×g, and 16,000×g for 30 min each. The supernatant and pellet were separated and analyzed by SDS–PAGE.

## Pulldown assay
For the structure-based mutagenesis studies, pulldown assays were performed with $Ni^{2+}$–NTA resins in 1 ml buffer containing 50 mM Tris, pH 7.5, 400 mM NaCl, 20 mM imidazole, and 0.1% Tween-20. His-tagged G6, SV2, or SV2–G6[AA] variants served as the baits and $H_CE$ (WT and variants) served as the prey. To prepare the pull-down, SV2 (5 μg) or SV2–G6[AA] (10 μg) were pre-incubated with $Ni^{2+}$–NTA resins at 12 °C for 1 h. After washing away the unbound proteins, the resins were mixed with $H_CE$ (32 μg, ~2-fold molar excess over the bait) at 12 °C for 1 h. The resins were then washed twice, and the bound proteins were released from the resins with 400 mM imidazole and subjected to SDS–PAGE.

To examine the interactions between $H_CE$ and SV2A–G6[AA] at various pH, we carried out the pull-down assays using Strep-Tactin resins (IBA Lifesciences) in three different buffers: 50 mM Tris, pH 7.5, 400 mM NaCl, and 0.1% Tween-20, or 50 mM sodium acetate, pH 5.0 or

4.6, 400 mM NaCl, and 0.1% Tween-20. The His-tagged SV2A–G6[AA] (10 μg) that was first biotinylated using EZ-Link NHS-PEG4-Biotin (Thermo Fisher Scientific) served as the bait and H$_C$E (32 μg) served as the prey. The pull-down assays were carried out at 12 °C for 1 h. The resins were then washed twice, and the bound proteins were released from the resins with 50 mM biotin and subjected to SDS–PAGE.

## Protein melting assay

The thermal stability of H$_C$E or SV2–G6[AA] variants was measured using a fluorescence-based thermal shift assay on a StepOne real-time PCR machine (Life Technologies). Each protein (~0.5 mg/ml) was mixed with the fluorescent dye SYPRO Orange (Sigma-Aldrich) and heated from 25 to 90 °C in a linear ramp. The midpoint of the protein-melting curve (Tm) was determined using the analysis software provided by the instrument manufacturer. Data obtained from three independent experiments were averaged to generate the bar graph.

## Biolayer interferometry assay

The binding between H$_C$E and SV2A–G6[AA], SV2A[Y535T/Y557D]–G6[AA], SV2C–G6[AA], and SV2C[T521Y/D543Y]–G6[AA] were examined by BLI assays using an OctetRED96 (ForteBio). Briefly, equal amounts of biotinylated SV2–G6[AA] variants (400 nM) were immobilized onto the Dip and Read Streptavidin (SA) Biosensors (ForteBio) and balanced with the buffer (50 mM Tris, pH 7.5, 400 mM NaCl, 0.5% BSA, and 0.1% Tween 20). The biosensors were then exposed to 1 μM H$_C$E (binding phase), followed by washing with the buffer (dissociation phase).

## Antibodies and constructs

The following antibodies were purchased from the indicated vendors: rabbit monoclonal antibodies against β-actin (ABclonal, AC038); mouse monoclonal antibodies against SNAP-25 (Synaptic systems, Cl 71.1) or Syt-1 (Synaptic systems, #105011); rabbit polyclonal antibody against SV2C (Synaptic systems, #119202). SV2 mouse monoclonal antibody (pan-SV2) was generously provided by E. Chapman (Madison, WI) and is available from Developmental Studies Hybridoma Bank (AB_2315387). Secondary antibodies were purchased from the following vendors: goat anti-rabbit-HRP (Bio-Rad, 1705046) and goat anti-mouse-HRP (Abcam, ab97023).

BoNT/E utilized for cell-based assays was purchased from Metabiologics or List Biologics (#141A) by the Dong lab. No recombinant BoNT/E was imported into the United States. All active BoNTs are stored in a locked freezer. Used toxins and contaminated media/reagents/containers are exposed to a 10% bleach solution for decontamination. Lentiviral constructs (in Lox-Syn-Syn vector) encoding full-length rat SV2A and SV2C were previously described[25]. Rat SV2Ac chimera was generated by replacing F487–E532 of SV2A with V473–K518 of SV2C using Gibson assembly and subcloned into Lox-Syn-Syn vector. SV2A (Y535T/Y557E) and SV2C (T521Y/E543Y) were generated by site-directed mutagenesis through overlapping PCR. All constructs were confirmed by sequencing (Genewiz).

## Mouse lines and pregnant rats

Sv2a- and Sv2b- knockout mice (strain B6;129P2-Sv2a[tm1Sud]Sv2b[tm1Sud]/J, stock No. 006383; cryo recovery) were obtained from the Jackson Laboratory. Mice heterozygous for both Sv2a and Sv2b were bred together to generate Sv2a+/− Sv2b−/− mice. Three primers were used to genotype Sv2a or 2b. Primers for Sv2a: mutant: GAG CGC GCG CGG CGG AGT TGT TGA C; wild type: GTT GAC TGA GAG TGA GAT GAG C; common: GAG TTA GGG ATG AGT GTT CTG G. Primers for Sv2b: mutant: GAG CGC GCG CGG CGG AGT TGT TGA C; wild type: TCA TCC AGA TGA TGT CAA GTC TAA GC; common: GGC ACT CAG CCA CTA ACT CTC AGT ACA). Once Sv2a+/− Sv2b−/− were established, they were bred to generate sv2a/sv2b double KO pups as mice homozygous for sv2a/sv2b double KO were not viable. Timed pregnant rats (Sprague Dawley strain) were purchased from Charles River.

## Neuron culture and lentivirus transduction

Rat cortical neurons were prepared from E18-19 embryos dissected from pregnant rats. Mouse Sv2a/2b double KO neurons were prepared from postnatal day 1 pup as previously described[12,27]. The pups were genotyped using Sv2a primers within 24 h after the pups were born. Sv2a−/−Sv2b−/− pups were used to culture the neurons. Dissected cortex was digested with papain for 1 h with tapping every 10–15 min, according to the manufacturer's instructions (Worthington Biochemical). Neurons were plated on poly-D-lysine-coated 24-well plates. Experiments were carried out generally with DIV (days in vitro) 13–15 neurons. Lentiviruses were prepared from HEK293T cells, as previously described[27]. 2.5 μM of arabinosylcytosine C (AraC) was added to neurons at DIV4, while lentiviruses were added at DIV5-6.

## H$_C$E binding to neurons

Neurons were exposed to 200 nM of biotinylated H$_C$E variants in high-K$^+$ buffer containing 87 mM NaCl, 56 mM KCl, 1.5 mM KH$_2$PO$_4$, 8 mM Na$_2$HPO$_4$, 0.5 mM MgCl$_2$, and 1 mM CaCl$_2$, for 5 min at 37 °C. Neurons were then washed three times with each 2.5 mL of phosphate-buffered saline (PBS). Neurons were harvested in a lysis buffer (PBS with 1% Triton X-100, 0.05% SDS, and protease inhibitor cocktail (Roche), 40 μL per well in 24-well plates). Lysates were centrifuged for 10 min at 4 °C, and the supernatants were subjected to SDS–PAGE and western blot analysis. Binding of H$_C$E was detected with Streptavidin-tagged horseradish peroxidase (HRP) (Cell signaling technology, 3999s), which recognizes biotinylated H$_C$E. All experiments were repeated three times independently.

## Entry of BoNT/E into neurons

BoNT/E was pre-activated with trypsin (Sigma, Type XIII-TPCK treated) for 30 min at 37 °C, and quenched with soybean trypsin inhibitor (Sigma, Type I-S, T6522, toxin:trypsin:inhibitor = 10:1:10 in molar ratio). Neurons were exposed to activated BoNT/E in the medium. Cells were incubated for 14–20 h at 37 °C. Neuron lysates were then harvested and subjected to western blot analysis to detect cleavage of SNAP-25 through chemiluminescence (SuperSignal West Pico Plus, Thermo Scientific).

## Mouse phrenic nerve hemidiaphragm assay

The MPN assay was performed employing 20–30 g Swiss mice (Janvier SA, France) as described previously[57]. Mice were euthanized by CO$_2$ anesthesia and subsequently exsanguinated. The phrenic nerve hemidiaphragm tissue was explanted, placed into an organ bath and continuously stimulated at 5–25 mA with a frequency of 1 Hz and a 0.1 ms pulse duration. Isometric contractions were transformed using a force transducer and recorded with VitroDat Online software (FMI GmbH, Germany). The time required to decrease the amplitude to 50% of the starting value (paralytic half-time) was determined. To allow comparison of the altered neurotoxicity of mutants with BoNT/E1 wild-type (displaying a specific activity of $0.41 \times 10^8$ LD$_{50}$/mg), its MPN assay dose-response-curve logarithmic function (y(BoNT/E1 wild-type; 2.0, 4.0, 8.0 pM) = −23.61 ln(x) + 104.11, $R^2$ = 0.999) consisting of three concentrations determined in 5–8 technical replicates as described previously[67] was employed. Mean (n = 3–5) of resulting paralytic half-times of the BoNT/E1 mutants were converted to concentrations of the wild-type employing the above function and finally expressed as relative neurotoxicity.

## Reporting summary

Further information on research design is available in the Nature Portfolio Reporting Summary linked to this article.

# Data availability

The coordinates and structure factors for the H$_C$E–G6 complex, the H$_C$E–SV2Ac–G6 complex, and the H$_C$E–SV2Ac–G6–sialic-acid complex have been deposited to the Protein Data Bank under accession codes

7UIE, 7UIA, and 7UIB, respectively. Source data are provided as a Source Data file. Other data supporting the findings of this study are available from the authors upon request. Other structures used in this study were obtained from the PDB with accession codes 2VU9 (H$_C$A–GT1b complex), 3BTA (BoNT/A), 3FFZ (BoNT/E), 3QUM (PSA–Fab complex), 5JLV (H$_C$A–SV2C complex), 6GLW (Fab fragment), 7OVW (GD1a-bound H$_C$E) Source data are provided with this paper.

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

## Acknowledgements

We would like to thank Dr. Feng Qiao for sharing with us the OctetRED96 (ForteBio) for BLI assays. This work was partly supported by National Institute of Health grants R21AI123920, R21AI156092, R01AI158503 to R.J.; R01AI125704 to R.J. and C.B.S.; R01AI139087 to R.J. and M.D.; R01NS080833 and R01AI132387 to M.D.; the German Federal Ministry of Education and Research grant 031L0111B/161L0111B to A.R. M.D. holds the Investigator in the Pathogenesis of Infectious Disease award from the Burroughs Wellcome Fund. NE-CAT at the Advanced Photon Source (APS) is supported by a grant from the National Institute of General Medical Sciences (P30 GM124165), and the Eiger 16M detector on the 24-ID-E beamline is funded by an NIH-ORIP HEI grant (S10OD021527). Use of the APS, an Office of Science User Facility operated for the U.S. Department of Energy (DOE) Office of Science by Argonne National Laboratory, was supported by the U.S. DOE under Contract No. DE-AC02-06CH11357.

## Author contributions

Conceptualization: Z.L. and R.J.; Methodology and Investigation: Z.L., P.-G.L., N.K., K.-h.L., H.L., A.P., P.C., G.Y., S.Z., and K.P.; Key reagents: J.M.T. and C.B.S.; Writing-original draft: Z.L. and R.J.; Writing-review and editing: Z.L., C.B.S., A.R., M.D., and R.J; Supervision: A.R., M.D., and R.J.; Funding acquisition: C.B.S., A.R., M.D., and R.J.

## Competing interests

R.J. is a co-founder, shareholder, and consultant of Claruvis Biotech, a shareholder of MingMed Biotech, and the founder of J&Z Consulting. R.J.'s relationship with these companies has been reviewed and approved by the University of California Irvine in accordance with its competing interest policies. A provisional patent application has been filed by The Regents of the University of California on the use of the structural information described in this manuscript to engineer BoNT/E for therapeutic and cosmetic applications with R.J. and Z.L. as inventors. The remaining authors declare no competing interests.
