## [Peer Review File · Nature Communications]

REVIEWER COMMENTS

Reviewer #1 (Remarks to the Author):

It is well known that botulinum neurotoxins (BoNTs) with different serotypes have very high specificity towards motor neurons due to their unique receptor binding mechanisms. In this manuscript, the authors report a robust piece of research describing the structural basis for botulinum neurotoxin E (BoNT/E) recognition of synaptic vesicle protein 2 using a combination of biochemical, structural and cell based techniques. This topic is of significant interest in the field, in particular as the toxin-receptor interactions amongst different serotypes seem to vary significantly and this manuscript does provide some important clues with BoNT/E which could be useful in the design of toxin based inhibitor molecules in the future.

The large amount of work reported here are of high quality and I would appreciate if the authors could answer the minor comments I have made below (authors' sections- first, followed by my queries) to clarify their detailed analysis.

Engineering a SV2A-2V2C chimera capable of recognizing HcE

•As SV2A-L4 and SV2C-L4 are homologous to each other, we sought to identify the HcE-binding region on SV2A-L4 and then replace the non-essential regions of SV2A L4 with the equivalent regions of SV2C-L4 in an effort to improve the biochemical behavior of SV2A-L4 while maintaining the SV2A-like binding with HcE

- How did the authors assess that the 'non' essential regions do not truly have an effect?**
- A large amount of work carried out to obtain an "SV2A"-like protein that co-crystalises with Hc/E but how much do the properties of SV2Ac differ to SV2A?**
- How do we know the linking of the VHH to SV2Ac won't have an effect on how it behaves?**
- Can we assume it will have an effect?**
- Is this effect of biological significance?**

Structure of HcE in complex with SV2A-

The peptide linker between SV2Ac and G6 had no visible electron density, indicating a highly flexible conformation. G6, in the context of SV2Ac-G6 fusion protein, bound HcE in the same manner as the stand alone G6, which further demonstrates that the peptide linker did not constrain SV2Ac and G6 association with HcE.

- Evidence for G6 not constraining SV2Ac, does this apply vice versa? Given that there is no stand-alone SV2Ac-Hc/E structure.**

BoNT/E selectively recognize SV2A and SV2B, but not SV2C

Discussion

Here, we determined the crystal structure of HcE in complex with a human SV2A

- Is this not misleading given that the authors actually determined the structure of Hc/E in complex with 'SV2Ac'?**

Methods

Crystallisation

o The authors need to include the original condition from which they optimised.

Figures

♣ In figure 1 the authors refer to it as SV2A, should this be SV2Ac?

Reviewer #2 (Remarks to the Author):

The authors of this study describe a structural and functional analysis of BoNT/E protein receptor binding, supporting a previous study by some of the authors that BoNT/E binds to SV2A and B but not C. The manuscript is well written and clearly organized. While the structural aspects of the study are scientifically sound and increase our understanding of the protein receptor-toxin interaction, the functional aspects and conclusions require additional analysis and discussion within current literature.

Major concerns:

1. A large portion of the manuscript is based on structures derived from an artificially created binding partner to HcE, the SV2A-G6AA. While an elegant solution to the problem of weak binding, it is not known or discussed whether the association of HcE with the VHH used in this approach can really mimic ganglioside binding. Can it be ruled out that G6AA binding might result in a conformational change of HcE that is different from that of ganglioside binding? In fact, the results described in lines 270-272 cast doubt on the validity of this approach. In addition, can it be ruled out that lipid association would further create a conformational change that might affect receptor binding?
2. The functional cell based assays do not seem to be conducted in a quantitative manner. These data cannot be interpreted without appropriate repeats and statistical analyses at the very least, but really should include dose-response curves. This is particularly true for Lines 312-321 and Fig 6b,c, where the small amount of SNAP-25 cleavage and differing SV2 expression levels as well as differing actin amounts (Fig 6c) make it impossible to interpret the data. In addition, the Western blot is over-exposed, so that differing levels in total SNAP-25 or actin would only be apparent if extreme. In addition, BoNT/A should be used as a control in this assay.
3. The BoNT/E was produced in E.coli, and specific activity not determined. It would be helpful to at least conduct an MPN or cell based assay assessing the specific activity of the BoNT/E used in the functional assays, ideally in comparison to C. botulinum produced BoNT/E, but at least compared to activity reported in the literature. The cell assay used 200 pM of the toxin and a 14-20 h exposure. The resulting SNAP-25 cleavage even when SV2A wt is expressed on the cells is surprisingly low.
4. The authors appropriately describe production of BoNT/E in Germany and associated biosecurity measures. However, it is unclear where the cell based assays were performed. If the recombinant mutated holotoxins were imported into the US and used in a US laboratory, a biosecurity statement must be provided.

Minor Concerns:

1. Lines 47: Actually, while it seems obvious that different neuronal cell entry strategies result in different pharmacological and clinical profiles, BoNT/A and /B also have different SNARE targets. Please consider revising 'contribute' to 'may contribute'
2. Line 54 and ref 25: Please consider softening this statement. There's other literature that is not consistent with this hypothesis.
3. Line 61: Please include newer reference showing flexibility ion the BoNT/A - SV2C binding site by Kammerer et al.

4. line 65: Consider removing 'designer' throughout manuscript
5. lines 74-75: This conclusion is going too far. these findings may contribute to the structural basis ... This statement is also assuming that in fact the protein receptor binding is a dominant factor that determines neuronal cell entry characteristics, which has not been proven in this ms.
6. line 126: 'G6AA successfully mimics the dual receptor binding ..' Is this a true statement? Binding to G6AA is via different residues and interactions than to gangliosides, so wouldn't this rather be a 'binding aide' to enable structural analysis than a 'mimic'?
7. lines 143 ff: Please add glycosylation status of constructs here. It becomes clear later in the text, but it would be helpful here.
8. line 153: remove 'faithfully'
9. Line 191: remove 'faithfully'
10. lines 234-236: Is it possible that this pocket might associated with complex gangliosides?
11. paragraph 238-254: This paragraph is confusing,, especially sentence 248-250. Should that sentence end with HcE? Please clarify
12. lines 276-292 and Fig 5b: It is a bit unclear whether the MPN shows a decrease in toxicity or a delay in toxicity, i.e. altered binding/entry kinetics. Please clarify.
13. Line 294: Again, this statement is considering only one citation by the authors involved in this ms, but other literature is not consistent with that hypothesis. Specifically, Restani L, Giribaldi F, Manich M, Bercsenyi K, Menendez G, Rossetto O, Caleo M, Schiavo G. Botulinum neurotoxins A and E undergo retrograde axonal transport in primary motor neurons. PLoS Pathog. 2012 Dec;8(12):e1003087.
and Pellett S, Tepp WH, Johnson EA. Botulinum neurotoxins A, B, C, E, and F preferentially enter cultured human motor neurons compared to other cultured human neuronal populations. FEBS Lett. 2019 Sep;593(18):2675-2685.
14. Line 336: this sentence needs a fig reference .
15. Line 351, ref 56: The authors worded this sentence appropriately, however, I would suggest to still add an additional explanation that the results in ref 56 using culture supernatants could be due to many factors, specific activity being just one.
- 16 line 372-382: Please see comment #13. This should be put into the context of all relevant literature. In addition, protein receptor binding may not be the limiting factor for BoNT/E entry.
17. line 427: Please add 'in Germany' after Dr. Rummel
18. Line 557: Please provide specific activity of BoNT/E, and the location of where this assay was conducted, and a biosecurity statement if needed
19. Fig 2c: SV2C-G6AA binding cannot be assessed in this figure, as it would be obscured by the large SV2A-G6AA band
20. Fig 2d: In addition to a statistical analysis of replicates and a dose-response analysis, this also seems like an unfair comparison. It should include over-expression of SV2C wt. In addition, it is unclear whether the KO cells still express SV2C. BoNT/A should be used as a control.
- 21: Line 804: Please consider re-wording the title of Fig 5 to better describe what aspect of BoNT/E the binding is crucial for.
22. Fig 5: Please provide dose-response data for the MPN as supplemental data. Please also include the wt as a bar in the graph.

Minor Concerns

Reviewer #3 (Remarks to the Author):

This is a well-conceived and executed study that takes advantage of a combination of different techniques to explore the structural underpinning of the recognition by BoNT/E of the distinct synaptic vesicle protein 2 (SV2) isoforms (SV2A, SV2B and SV2C). The crystal structure of a 3-component complex composed of a single-domain camelid antibody that acts as a ganglioside binding surrogate on the receptor binding domain of BoNT/E (HCE), a fusion protein (SV2Ac) encompassing key sequences of the receptor proteins SV2A and SV2C, and HCE affords recognition of the determinants for the receptor specificity of BoNT/E in contrast to BoNT/A. Figure 3 highlights a key result discerning the two components on SV2 recognized by BoNT/E, namely a protein-protein interface and an N-glycan binding cleft on two distant sites. These are striking features of the structure that outline the basis for the distinct recognition of BoNT/E of SV2A and SV2B but not SV2C, the latter being recognized by BoNT/A yet not by BoNT/E. Given the overall structural similarity of BoNT/E and BoNT/A the authors reasonably assert that while BoNT/A recognizes all three isoforms of SV2 based on backbone-backbone interactions (not sequence), BoNT/E discerns small sequence differences between SV2A and SV2C, combined with different protein-glycan binding modalities of BoNT/A and BoNT/E.

The study is a significant and timely advance to elucidate the molecular basis of the receptor specificity differences between BoNT/E and BoNT/A. As such, the report merits publication after the authors consider revisions that may improve the impact of the new results.

Comments:

The last paragraph appears uncommitted. The results allow the authors to be more specific about BoNT structural design, drug design and even antibody prevention strategies. Such additions may provide more excitement to the presentation without deteriorating into the category of speculative.

The structure and the supplementary results do not offer insights into the underlying mechanism for the faster onset of action and the shorter duration of action of BoNT/E relative to BoNT/A. This is a crucial issue that merits expansion and that the authors may wish to address.

Minor points:

The results on the pH dependence of HCE binding of SV2A are interesting and intriguing. The authors may wish to elaborate on the mechanism of dissociation of HC from HN inside the acidic environment of endosomes.

Define HCCE and HCNE earlier in the text

Notation of the chimera may be improved to avoid confusion. Perhaps using italics SV2Ac.

Complete check of the references for accuracy and completeness, for example:

Reference 6, incomplete

Reference 31, author misspelled (R. Jahn)

Reference 61 -Kabsch W, X ds ?

Structural basis for botulinum neurotoxin E recognition of synaptic vesicle protein 2

We thank the editor and the reviewers for their careful reading of the manuscript and their constructive suggestions that have guided our revision. We have revised the manuscript to address reviewers' concerns as outlined below. We sincerely apologize for the long delay in finishing this revision due to our difficulty in generating sufficient numbers of SV2 knockout mice during this period.

Reviewer #1

It is well known that botulinum neurotoxins (BoNTs) with different serotypes have very high specificity towards motor neurons due to their unique receptor binding mechanisms. In this manuscript, the authors report a robust piece of research describing the structural basis for botulinum neurotoxin E (BoNT/E) recognition of synaptic vesicle protein 2 using a combination of biochemical, structural and cell based techniques. This topic is of significant interest in the field, in particular as the toxin-receptor interactions amongst different serotypes seem to vary significantly and this manuscript does provide some important clues with BoNT/E which could be useful in the design of toxin based inhibitor molecules in the future.

The large amount of work reported here are of high quality and I would appreciate if the authors could answer the minor comments I have made below (authors' sections- first, followed by my queries) to clarify their detailed analysis.

Engineering a SV2A-2V2C chimera capable of recognizing HcE

•As SV2A-L4 and SV2C-L4 are homologous to each other, we sought to identify the HCE-binding region on SV2A-L4 and then replace the non-essential regions of SV2A L4 with the equivalent regions of SV2C-L4 in an effort to improve the biochemical behavior of SV2A-L4 while maintaining the SV2A-like binding with HCE

- How did the authors assess that the 'non' essential regions do not truly have an effect?*
- A large amount of work carried out to obtain an "SV2A"-like protein that co-crystalises with Hc/E but how much do the properties of SV2Ac differ to SV2A?*
- How do we know the linking of the VHH to SV2Ac won't have an effect on how it behaves?*
- Can we assume it will have an effect?*
- Is this effect of biological significance?*

Response: We have reworded this sentence as our purpose was not to identify the essential or non-essential region of SV2A. It now reads as "As SV2A-L4 and SV2C-L4 are homologous to each other, we sought to develop a SV2A-SV2C chimera that has improved biochemical behavior over SV2A-L4 while maintaining the SV2A-like binding with HcE" (lines 143-145). Our structure reveals that most of the HcE-interacting residues are native SV2A residues located on the SV2A portion of the chimera. Only two HcE-binding amino acids are located on the SV2C portion of the chimera, which are G500 and R501 on SV2C, equivalent to D514 and K515 on SV2A, respectively. We confirmed that the two SV2A-like residues at these positions are able to maintain HcE binding. Therefore, the crystal structure shows that SV2Ac chimera faithfully mimics SV2A in terms of HcE binding. Furthermore, we carried out the following studies to validate the structural findings. (1) We examined the interactions between a series of HcE mutants that we designed based on the crystal structure and SV2A-G6^{AA} using *in vitro* pull-down

assays (Supplementary Fig. 3e). Please note that the pull-down studies were carried out using the WT SV2A, but not the SV2Ac fusion protein. (2) We examined how these HcE mutants recognized endogenous SV2A and SV2B in cultured rat cortical neuron (Fig. 5a). (3) The accuracy of the structural findings was further validated by the MPN assay (Fig. 5b). (4) The structure helped us identify signature amino acids on SV2 that are involved in different recognition of HcE towards SV2A/SV2B versus SV2C, which was validated by the design and characterization of a loss-of-function SV2A mutant and a gain-of-function SV2C mutant on neurons (Fig. 6b-c).

We think that the linking of G6 to SV2Ac via a flexible peptide unlikely had an effect on SV2Ac binding based on the following reasons. (1) G6 in the context of SV2Ac–G6 fusion protein bound HcE in the same manner as the stand alone G6, suggesting G6 was not constrained by SV2Ac or the peptide linker. (2) The peptide linker was invisible in the crystal structure, indicating that it has a flexible conformation that is not constrained by the two proteins. (3) We purposely designed G6^{AA} to largely tune down its binding affinity to HcE in order not to mask the weak binding contributed by SV2Ac during design and characterization. (4) Neither G6^{AA} nor SV2Ac could pull down HcE by itself, while the fusion of two showed robust binding to HcE. The clear binding synergy between G6^{AA} and SV2Ac in the fusion protein suggests that both of them were able to bind properly albeit weakly.

Structure of HcE in complex with SV2A-

The peptide linker between SV2Ac and G6 had no visible electron density, indicating a highly flexible conformation. G6, in the context of SV2Ac–G6 fusion protein, bound HcE in the same manner as the stand alone G6, which further demonstrates that the peptide linker did not constrain SV2Ac and G6 association with HcE.

• Evidence for G6 not constraining SV2Ac, does this apply vice versa? Given that there is no stand-alone SV2Ac-Hc/E structure.

Response: This is related to the first question. The constraining effect, if any, should have mutual effect on both proteins. Given the very weak binding affinities of HcE to SV2Ac or G6^{AA} separately, if G6 was to constrain SV2Ac binding or vice versa, the SV2Ac–G6^{AA} fusion protein should not have bound HcE robustly as what we observed. Furthermore, our structural findings have been validated by systematic mutagenesis studies.

BoNT/E selectively recognize SV2A and SV2B, but not SV2C

Discussion

Here, we determined the crystal structure of HcE in complex with a human SV2A

• Is this not misleading given that the authors actually determined the structure of Hc/E in complex with ‘SV2Ac’?

Response: We made the change as suggested (lines 340-341).

Methods, Crystallisation

o The authors need to include the original condition from which they optimised.

Response: We included this information in the Method (lines 458-461).

Figures ♣ In figure 1 the authors refer to it as SV2A, should this be SV2Ac?

Response: This is in fact the WT SV2A. The WT SV2A and its fusion with G6 formed soluble aggregates in solution, which prevented them from being crystallized, but was good enough for the pull down assays here as well as that shown in Fig. 2c and Supplementary Fig. 3e-f.

Reviewer #2

The authors of this study describe a structural and functional analysis of BoNT/E protein receptor binding, supporting a previous study by some of the authors that BoNT/E binds to SV2A and B but not C. The manuscript is well written and clearly organized. While the structural aspects of the study are scientifically sound and increase our understanding of the protein receptor-toxin interaction, the functional aspects and conclusions require additional analysis and discussion within current literature.

Major concerns:

1. A large portion of the manuscript is based on structures derived from an artificially created binding partner to HcE, the SV2A-G6AA. While an elegant solution to the problem of weak binding, it is not known or discussed whether the association of HcE with the VHH used in this approach can really mimic ganglioside binding. Can it be ruled out that G6AA binding might result in a conformational change of HcE that is different from that of ganglioside binding? In fact, the results described in lines 270-272 cast doubt on the validity of this approach. In addition, can it be ruled out that lipid association would further create a conformational change that might affect receptor binding?

Response: In fact, the structure of HcE in complex with GD1a was recently reported by the Stenmark group (ref. 48). The superimposed structures of HcE-GD1a and HcE-G6 complexes are shown in Fig. 1d. Our analyses of both structures demonstrate that (1) G6 binding did not result in a conformational change of HcE in comparison to the GD1a-bound HcE (r.m.s.d. is ~ 0.38 Å over 359 superimposed amino acids); and (2) GD1a binding does not change the overall structure of HcE excepts for rearrangement of a loop 1228–1237 that is suggested to be triggered by GD1a (ref. 48). But this loop is located far from the SV2A-binding site and will not affect SV2A binding. We have added some of these discussions to the revised manuscript (lines 119-120).

The reason we did not observe detectable change of binding between HcE^{Y879G/Y881G} or HcE^{Y891G/Y1041G} and SV2A-G6^{AA} *in vitro* using pull-down assays is likely due to the heterogeneous glycosylation of the recombinant SV2A. Therefore, the *in vitro* pull-down assay is mostly detecting the protein-mediated interactions, while the neuron binding of HcE and intoxication by full-length BoNT/E is mediated by both protein- and glycan-based interactions. This is why we carried out three independent binding studies using recombinant proteins, neurons, and the MPN assay in order to complement each other.

2. The functional cell based assays do not seem to be conducted in a quantitative manner. These data cannot be interpreted without appropriate repeats and statistical analyses at the very least, but really

should include dose-response curves. This is particularly true for Lines 312-321 and Fig 6b,c, where the small amount of SNAP-25 cleavage and differing SV2 expression levels as well as differing actin amounts (Fig 6c) make it impossible to interpret the data. In addition, the Western blot is over-exposed, so that differing levels in total SNAP-25 or actin would only be apparent if extreme. In addition, BoNT/A should be used as a control in this assay.

Response: We thank the reviewer for the suggestion and agree that it would be ideal to have dose-response curves. We have been repeating these experiments in the past year for this purpose, but this has been very challenging due to low availability of SV2 knock out (KO) neurons and variability in cell numbers/conditions when culturing them from different mouse pups.

Cortical neurons mainly express SV2A and SV2B. As SV2A KO mice cannot survive beyond 21 days, we first have to breed mice to generate SV2A^{+/+}-SV2B^{-/-}. Breeding pairs of SV2A^{+/+}-SV2B^{-/-} generates pups with SV2A^{-/-}-SV2B^{-/-} genotype with a chance of only 25%. In addition, these double KO mice have a lower survival rate compared with their littermates. We have to dissect the pups within 24 hours after they are born, and this has been difficult during reduced capacity in the animal facility. Thus, we often only obtain one or limited numbers of double KO pups, which each time can only provide limited numbers of cells for western blot purposes. These limitations prevented us from having proper dose-response curves in the previous manuscript, and we apologize for this shortcoming of our study.

In the revised manuscript, we replaced the data in the original Fig. 6b with three toxin doses and the cleavage of SNAP-25 was quantified and plotted based on the comparison between the intact and the cleaved SNAP-25 (new Fig. 6b). These results are consistent with the previous Fig. 6b and confirmed that SV2A^{Y535T/Y557E} cannot function as a receptor for BoNT/E at all these three doses.

For Fig. 6c, we were able to update the data with two new toxin doses and both showed the same results as the original Fig. 6c. In all three conditions, we only observed a relatively minor cleavage of SNAP-25 on neurons expressing the mutated SV2C^{T521Y/E543Y} despite the dose differences. Therefore, there is no need to quantify them. In comparison, neurons expressing wild type (WT) SV2C did not show cleavage of SNAP-25 under the same conditions. These data suggest that SV2C^{T521Y/E543Y} indeed gained the capability as a receptor for BoNT/E, although this capability is rather limited. Additional mutations might be necessary to further enhance recognition by BoNT/E. As suggested by the reviewer, we also tested BoNT/A as a control, which showed that both SV2C WT and SV2C^{T521Y/E543Y} can mediate entry of BoNT/A in neurons (new Supplementary Fig. 8). Taken together, we believe that our limited functional assays at semiquantitative levels, although less than ideal, are sufficient to validate the structural approach and structural findings, which are the main purpose of this manuscript.

3. The BoNT/E was produced in E.coli, and specific activity not determined. It would be helpful to at least conduct an MPN or cell based assay assessing the specific activity of the BoNT/E used in the functional assays, ideally in comparison to C. botulinum produced BoNT/E, but at least compared to activity reported in the literature. The cell assay used 200 pM of the toxin and a 14-20 h exposure. The resulting SNAP-25 cleavage even when SV2A wt is expressed on the cells is surprisingly low.

Response: We clarify that all BoNT/E utilized in cell-based assays was BoNT/E3 purchased from either Metabionics or List Biologics (during the revision due to limited availability from Metabionics). The toxin was activated using trypsin treatment. No recombinant BoNT/E was imported into the US. We added this information in the Method section. To investigate BoNT/E mutants, recombinantly expressed BoNT/E1 was utilized in the Rummel lab in Germany. The wild-type BoNT/E1, its MPN assay dose-

response curve function (y (BoNT/E; 2.04/4.08/8.15 pM) = $-23.612\ln(x) + 104.11$, $R^2 = 1.0$) used for calculating the neurotoxicity of the BoNT/E mutants and the specific activity of 0.41×10^8 LD₅₀/mg were described previously in Weisemann et al. Toxins 2015 (ref. 67). This dose-response curve function was already mentioned in the Method section and the specific activity data has been added to the manuscript (new Supplementary Fig. 6).

4. The authors appropriately describe production of BoNT/E in Germany and associated biosecurity measures. However, it is unclear where the cell based assays were performed. If the recombinant mutated holotoxins were imported into the US and used in a US laboratory, a biosecurity statement must be provided.

Response: The BoNT/E used in Dr. Dong's lab was purchased from Metabionics and List Biologics. The recombinantly expressed WT and the mutated BoNT/E holotoxin were not imported into the US and not used in a US laboratory.

Minor Concerns:

1. Lines 47: Actually, while it seems obvious that different neuronal cell entry strategies result in different pharmacological and clinical profiles, BoNT/A and /B also have different SNARE targets. Please consider revising 'contribute' to 'may contribute'

Response: We made the change as suggested.

2. Line 54 and ref 25: Please consider softening this statement. There's other literature that is not consistent with this hypothesis.

13. Line 294: Again, this statement is considering only one citation by the authors involved in this ms, but other literature is not consistent with that hypothesis. Specifically, Restani L, Giribaldi F, Manich M, Bercsenyi K, Menendez G, Rossetto O, Caleo M, Schiavo G. Botulinum neurotoxins A and E undergo retrograde axonal transport in primary motor neurons. PLoS Pathog. 2012 Dec;8(12):e1003087. and Pellett S, Tepp WH, Johnson EA. Botulinum neurotoxins A, B, C, E, and F preferentially enter cultured human motor neurons compared to other cultured human neuronal populations. FEBS Lett. 2019 Sep;593(18):2675-2685.

16 line 372-382: Please see comment #13. This should be put into the context of all relevant literature. In addition, protein receptor binding may not be the limiting factor for BoNT/E entry.

Responses: We thank the reviewer for this suggestion, and we softened these statements and cited these two references (refs. 31-32) (lines 54-56, 386-389). We also added a new ref #26 (Mahrhold et al. 2013) that demonstrates that HcE cannot bind to N-glycosylated SV2C-L4 whereas HcA binds N-glycosylated SV2C-L4 and N-glycosylated SV2A-L4. Our previous evidence is limited to expression of SV2C in cortical neurons and in vitro biochemical assays. We agree with the reviewer that whether SV2C in motor neurons can be a receptor has not been excluded. We also agree that protein receptor binding, although critical for the potency, may not be the limiting factor.

3. Line 61: Please include newer reference showing flexibility ion the BoNT/A - SV2C binding site by Kammerer et al.

Response: We added three additional references here (new ref. #36-38).

4. line 65: Consider removing 'designer' throughout manuscript.

Response: Removed.

5. lines 74-75: This conclusion is going too far. these findings may contribute to the structural basis ... This statement is also assuming that in fact the protein receptor binding is a dominant factor that determines neuronal cell entry characteristics, which has not been proven in this ms.

Response: We did not claim that the protein receptor is a dominant factor determining neuronal entry in this manuscript. But the importance of protein receptor is evidenced by the observation that BoNT/B displays significantly lower potency on human than mouse, which is largely due to a singly amino acid difference on human and mouse synaptotagmin. The recent structure-based engineering of BoNT/B has successfully enhanced its binding to human synaptotagmin and led to improved clinical efficacy. Our studies reported here will provide the structural basis for BoNT/E engineering in future studies.

6. line 126: 'G6AA successfully mimics the dual receptor binding ..' Is this a true statement? Binding to G6AA is via different residues and interactions than to gangliosides, so wouldn't this rather be a 'binding aide' to enable structural analysis than a 'mimic'?

Response: We agree that the wild-type G6 is serving like a “binding aide” during structure analysis as it binds to HcE with a high affinity. But mechanism wise, G6^{AA} does play like a ganglioside mimic because it has very weak binding to HcE by itself due to the two AA mutations, while SV2Ac–G6^{AA} robustly binds to HcE in a sense mimicking simultaneous binding of SV2 and ganglioside on neuron surface.

7. lines 143 ff: Please add glycosylation status of constructs here. It becomes clear later in the text, but it would be helpful here.

Response: These SV2A-SV2C chimeras as well as all the SV2A–G6 fusion proteins were made in HEK cells and glycosylated. We clarified this in the text (lines 132 and 147).

8. line 153: remove 'faithfully'

9. Line 191: remove 'faithfully'

Response: Done.

10. lines 234-236: *Is it possible that this pocket might associated with complex gangliosides?*

Response: It is reasonable to hypothesize that this pocket on H_CE may be involved in recognition of sialic acids on other neuronal glycoproteins or glycolipids. This possibility was mentioned in the Discussion (lines 355-357).

11. paragraph 238-254: *This paragraph is confusing, especially sentence 248-250. Should that sentence end with H_CE? Please clarify.*

Response: In this section, we tried to compare two different glycan-binding modes between H_CE and H_CA. The main point is that the glycan-binding modes are largely determined by the protein-protein binding modes between H_C and SV2: (1) for H_CA, the N-glycan is conjugated to N559 of SV2C, which allows H_CA to interact with its glycan core, while its distal sialic acid is located too far to be bound; (2) For H_CE, the N-glycan is conjugated to N573 of SV2A, and in this binding mode, the glycan core is located too far to bind H_CE, while its distal sialic acid could reach the glycan-binding pocket on H_CE. We rewrote this paragraph to clarify this (lines 246-260).

12. lines 276-292 and Fig 5b: *It is a bit unclear whether the MPN shows a decrease in toxicity or a delay in toxicity, i.e. altered binding/entry kinetics. Please clarify.*

Response: The paralytic half times measured in MPN assay for BoNT/E wild-type show a clear dose-dependence. Lower toxin dose corresponds to longer paralytic half time and lower toxicity since less endplates are intoxicated simultaneously with less toxin molecules. Applying mutants with impaired receptor binding, less BoNT molecules will bind to their receptors and less BoNT molecules are taken up mimicking a lower dose of BoNT/E wild-type thereby displaying reduced neurotoxicity. Impaired receptor binding of BoNT/E mutants can be compensated by increasing applied concentrations of BoNT/E mutants to increase number of bound BoNT/E mutants to increase number of intoxicated endplates. Whether the onset of symptoms by BoNT/E mutants is changed upon impairing the protein receptor binding site needs to be demonstrated in vivo e.g. in DAS assays.

14. Line 336: *this sentence needs a fig reference.*

Response: We now cited Fig. 5a-b and Supplement Fig. 3e (line 348).

15. Line 351, ref 56: *The authors worded this sentence appropriately, however, I would suggest to still add an additional explanation that the results in ref 56 using culture supernatants could be due to many factors, specific activity being just one.*

Response: We revised the sentence as follows, “For example, a key SV2A-binding residue R1100 is replaced with S1100 on subtype BoNT/E10 and E11, which could weaken receptor binding and may be partly responsible for the reported lower toxicity of culture supernatants containing BoNT/E10 and E11 besides other factors such as growth rate and toxin secretion⁶³ (Supplementary Fig.9).” (lines 362-366)

17. line 427: Please add 'in Germany' after Dr. Rummel

Response: Added.

18. Line 557: Please provide specific activity of BoNT/E, and the location of where this assay was conducted, and a biosecurity statement if needed

Response: We clarified that all cell-based assays were carried out in the Dong lab using native BoNT/E purchased from either Metabionics (original figures) or List Biologics (during revision). These toxins are kept under exempted amount in the lab, no biosecurity statement is needed.

19. Fig 2c: SV2C-G6AA binding cannot be assessed in this figure, as it would be obscured by the large SV2A-G6AA band.

Response: We agreed that this was not an ideal situation since the band of HcE and the smear bands of the glycosylated SV2 and SV2-G6 happened to be partially overlapping. However, this figure was not meant for any quantification purpose. We believe Fig. 2c is sufficient to show that SV2Ac-G6^{AA} and SV2A-G6^{AA} could pull down HcE, while the other proteins “did not show detectable binding with HcE in this assay”.

20. Fig 2d: In addition to a statistical analysis of replicates and a dose-response analysis, this also seems like an unfair comparison. It should include over-expression of SV2C wt. In addition, it is unclear whether the KO cells still express SV2C. BoNT/A should be used as a control.

Response: Please see our explanation of technical limitation in response to question #2 (Reviewer 2). Briefly, we were able to repeat this experiment with two more different doses. In both cases, we observed similar levels of cleavage of SNAP-25 between SV2A and SV2Ac. These results are consistent with our previous Fig. 2e and confirmed that SV2Ac could serve as a receptor for BoNT/E in neurons. We also tested BoNT/A as a control, which showed that both SV2A and SV2Ac can mediate entry of BoNT/A (new Supplementary Fig. 2). BoNT/A did not enter SV2 KO cells, confirming that these neurons do not express SV2C. Over-expression of SV2C is addressed in the new Fig. 6c.

21: Line 804: Please consider re-wording the title of Fig 5 to better describe what aspect of BoNT/E the binding is crucial for.

Response: We changed the title of Fig. 5 to “Simultaneous binding to the protein- and glycan-moiety of SV2A is crucial for neuron binding and neurotoxicity of BoNT/E”.

22. Fig 5: Please provide dose-response data for the MPN as supplemental data. Please also include the

wt as a bar in the graph.

Response: BoNT/E wt data was added to the modified Fig. 5b and the new Supplementary Fig. 6 shows the dose-response-curve data of BoNT/E wild-type.

Reviewer #3

This is a well-conceived and executed study that takes advantage of a combination of different techniques to explore the structural underpinning of the recognition by BoNT/E of the distinct synaptic vesicle protein 2 (SV2) isoforms (SV2A, SV2B and SV2C). The crystal structure of a 3-component complex composed of a single-domain camelid antibody that acts as a ganglioside binding surrogate on the receptor binding domain of BoNT/E (HCE), a fusion protein (SV2Ac) encompassing key sequences of the receptor proteins SV2A and SV2C, and HCE affords recognition of the determinants for the receptor specificity of BoNT/E in contrast to BoNT/A. Figure 3 highlights a key result discerning the two components on SV2 recognized by BoNT/E, namely a protein-protein interface and an N-glycan binding cleft on two distant sites. These are striking features of the structure that outline the basis for the distinct recognition of BoNT/E of SV2A and SV2B but not SV2C, the latter being recognized by BoNT/A yet not by BoNT/E. Given the overall structural similarity of BoNT/E and BoNT/A the authors reasonably assert that while BoNT/A recognizes all three isoforms of SV2 based on backbone-backbone interactions (not sequence), BoNT/E discerns small sequence differences between SV2A and SV2C, combined with different protein-glycan binding modalities of BoNT/A and BoNT/E.

The study is a significant and timely advance to elucidate the molecular basis of the receptor specificity differences between BoNT/E and BoNT/A. As such, the report merits publication after the authors consider revisions that may improve the impact of the new results.

Comments:

The last paragraph appears uncommitted. The results allow the authors to be more specific about BoNT structural design, drug design and even antibody prevention strategies. Such additions may provide more excitement to the presentation without deteriorating into the category of speculative.

Response: We thank the reviewer for your enthusiasm on our studies and the opportunities for BoNT engineering based on the new structure. But we were hesitant to make specific suggestions without experimental verification, and we hope more labs will join us to explore these opportunities upon timely publication of this study.

The structure and the supplementary results do not offer insights into the underlying mechanism for the faster onset of action and the shorter duration of action of BoNT/E relative to BoNT/A. This is a crucial issue that merits expansion and that the authors may wish to address.

Response: We agree that our current studies do not offer direct insights into this unique feature of BoNT/E. It is generally believed that the shorter duration of BoNT/E is due to the shorter lifetime of LC/E than LC/A in the cytosol. The mechanism underlying faster onset of BoNT/E remain mysterious, which has many contributing factors including receptor binding, pH-dependent conformational changes,

LC unfolding and refolding, pore formation, and membrane translation.

Minor points:

The results on the pH dependence of HCE binding of SV2A are interesting and intriguing. The authors may wish to elaborate on the mechanism of dissociation of HC from HN inside the acidic environment of endosomes.

Response: At this moment, we still don't quite understand the mechanism underlying the pH-dependent binding between HcE and SV2A, which would be an intriguing question for future studies.

Define HCCE and HCNE earlier in the text

Response: We now define them in the first section of the text (line 104) and the legend of Fig. 1.

Notation of the chimera may be improved to avoid confusion. Perhaps using italics SV2Ac.

Response: We are afraid that an italic SV2Ac may be misunderstood as a gene name.

Complete check of the references for accuracy and completeness, for example:

Reference 6, incomplete

Reference 31, author misspelled (R. Jahn)

Reference 61 -Kabsch W, Xds ?

Response: Thanks for your suggestions: (1) Ref. 6 is fixed; (2) Ref. 31 was from Roger Janz and Südhof; (3) Ref. 61 is a bit confusing as "XDS" is the title of the paper that is a computing software we used for processing X-ray diffraction data.

REVIEWERS' COMMENTS

Reviewer #1 (Remarks to the Author):

The authors have addressed my queries in the revised version with adequate details. The new version reads well. A nice piece of work worthy of publication in this journal.

Reviewer #2 (Remarks to the Author):

The authors have gone to great length to address all concerns, which they did expertly. The study will make an important contribution to the field and I recommend publication